# Efficient and Uncertainty-Aware Diffusion Framework for Offline-to-Online Reinforcement Learning

**Ha Manh Bui** [1]  **Metod Jazbec** [2]  **Eric Nalisnick** [1]  **Anqi Liu** [1]

## Abstract

Offline-to-Online Reinforcement Learning (O2O-RL) leverages an offline, pre-trained policy to minimize costly online interactions. Although data-efficient, O2O-RL is susceptible to shifts between offline and online distributions. Existing work aims to mitigate the harm of this shift by finetuning the policy on trajectory data sampled from a diffusion model. Inspired by this line of work, we propose DUAL: an efficient **D**iffusion **U**ncertainty-**A**ware framework for offline-to-online reinforcement **L**earning. DUAL utilizes the prior knowledge of the diffusion model to distill a fast-sampling diffusion actor policy and transition model in the offline phase. DUAL also employs a Laplace approximation and distance transition-state-shift detection, thereby using uncertainty quantification to improve exploration versus exploitation in the online phase. We formally show that our actor loss with the Laplace approximation provides a proxy for a principled estimate of epistemic uncertainty. Empirically, DUAL improves the online expected return over O2O-RL baselines across multiple settings and environments.

## 1. Introduction

Online Reinforcement Learning (RL) often requires extensive interactions of an agent with the environment to learn the optimal policy, a process that can be costly in high-stakes applications. In contrast, offline RL utilizes pre-existing datasets to train a policy, avoiding the need for resource-intensive online interactions. Hence, to achieve optimal policy with fewer interactions in the online RL phase,

Offline-to-Online RL (O2O-RL) has emerged as a promising approach in the real world (Lee et al., 2022; Sutton and Barto, 2018). In O2O-RL, the agent undergoes two phases of learning. First, the policy is pre-trained from an offline dataset. Then, this policy is used to interact with the environment and fine-tuned through online interactions. However, due to the shift between online and offline distributions (Yu and Zhang, 2023; Lee et al., 2022), most O2O-RL algorithms (Kostrikov et al., 2022; Nakamoto et al., 2023; Zhang et al., 2024) still need several costly online interactions to learn the optimal policy in the online phase.

Capitalizing on diffusion models' strength in modeling complex distributions over large offline datasets (Ho et al., 2020), recent work has extensively applied them to offline RL (Zhu et al., 2024b;a; Psenka et al., 2024; Li et al., 2023; Ni et al., 2023; Lee et al., 2024; Wang et al., 2026). In general, there are two popular usages of diffusion models in RL, including *diffusion planner* and *diffusion policy*. The diffusion planner (Janner et al., 2022; Liang et al., 2023; Foffano et al., 2026) applies diffusion models to planning sequential decisions (i.e., the whole trajectory) with a sequence of state and action pairs. In O2O-RL, to address distribution shift, Liu et al. (2024); Huang et al. (2025) have recently pre-trained a diffusion planner on offline datasets, then generated trajectories aligned with the online distribution, thereby fine-tuning actor-critic models with augmented data. However, this kind of approach does not fully exploit diffusion models' modeling capabilities beyond data augmentation, and their policy learning still relies on simple function classes.

On the other hand, diffusion policy (Wang et al., 2023b; Ding et al., 2024; Ying et al., 2026) uses diffusion models as the parameterized policy class to output a single-step action, where the sampling target is the action conditioned on the state. Empirically, diffusion policy is often simple and performs well on short-term planning tasks (e.g., controlling a robot to maximize movement speed) by leveraging diffusion models' ability to model complex action distributions. That said, the diffusion policy, which operates without lookahead planning, often underperforms diffusion planner in complex long-term planning RL tasks (e.g., robot navigation, manipulation tasks) (Lu et al., 2025; Chen et al., 2023; Scheikl et al., 2024). When used in a policy training and inference loop

[1]Department of Computer Science, Johns Hopkins University, Baltimore, MD, U.S.A. [2]AMLab, University of Amsterdam, Amsterdam, Netherlands. Correspondence to: Ha Manh Bui <hb.buimanhha@gmail.com>.

*Proceedings of the 43rd International Conference on Machine Learning*, Seoul, South Korea. PMLR 306, 2026. Copyright 2026 by the author(s).

(e.g., Diff-QL (Wang et al., 2023b), IDQL (Hansen-Estruch et al., 2023), DTQL (Chen et al., 2024), DACER (Wang et al., 2024), etc.), there are also concerns about the trade-off between computational speed and output quality, via the choice of the number of denoising steps. In addition, in the diffusion for O2O-RL (Liu et al., 2024; Huang et al., 2025), the policy, either deterministic or parameterized as a distribution with learned variance, only provide aleatoric (data) uncertainty, and is often overconfident under distribution shifts (Chua et al., 2018; Manh Bui and Liu, 2024). In O2O-RL, without epistemic (model) uncertainty, such overconfident models usually exploit a suboptimal policy, thereby hindering exploration of the optimal policy in the online phase (Osband et al., 2016; Mnih et al., 2015).

Motivated by both diffusion planner and diffusion policy, we ask whether we can exploit the benefit of both sides in a principled way for O2O-RL. Specifically, we ask the following two questions: *(1) Can we design an efficient diffusion actor policy that inherits diffusion planner's long-term planning ability in O2O-RL?; (2) Can we improve the UQ of this diffusion policy to balance exploration and exploitation in the online phase?*

To answer the two questions above, we introduce DUAL, a unified diffusion framework for O2O-RL. Specifically, DUAL utilizes the prior knowledge of diffusion planner to distill a fast-sampling diffusion actor policy and transition model in the offline phase. This helps DUAL's policy training and quick inference while maintaining high output quality by leveraging the rich expressiveness of the diffusion planner. After that, we design a novel diffusion-policy UQ method that includes two terms. The first is a theoretically sound epistemic uncertainty term by the Laplace approximation with the policy improvement in actor-critic. The second term is a distance transition-state-shift detection by the extracted diffusion-based transition model. These two novel terms enable DUAL to account for model uncertainty and measure distribution shifts, thereby supporting the balance between exploration and exploitation in O2O-RL. Our contributions can be summarized as follows:

1. We rigorously introduce DUAL: an efficient **D**iffusion **U**ncertainty-**A**ware framework for offline-to-online reinforcement **L**earning. Notably, DUAL is compatible with and can be implemented on top of multiple modified critics and data-augmentation O2O-RL related work.

2. We formally show that DUAL's actor loss with the Laplace approximation provides a principled estimate of epistemic uncertainty. This helps DUAL leverage a high-quality epistemic uncertainty to balance exploration and exploitation in the online phase.

3. We empirically show DUAL is computationally efficient and achieves a notable online return improvement over O2O-RL and diffusion-RL baselines across MuJoCo,

AntMaze, Frozen-Lake, and Adroit environments. Ablation studies confirm the importance of DUAL's UQ to learn the optimal policy in online phase.

## 2. Preliminary

We consider a Markov Decision Process (MDP): $\mathcal{M}_\gamma := (\mathcal{S}, \mathcal{A}, \mathbb{P}, r, \gamma, \rho_0)$, where $\mathcal{S}$ is the state space, $\mathcal{A}$ is the action space, $\mathbb{P} : \mathcal{S} \times \mathcal{A} \to \Delta(\mathcal{S})$ is the transition operator that takes a state-action pair and returns a distribution over states, $r : \mathcal{S} \times \mathcal{A} \to [0, R_{\max}]$ is the deterministic reward function, $\gamma \in [0, 1)$ is the discount factor, and $\rho_0$ is the initial distribution over states. A policy $\pi : \mathcal{S} \to \Delta(\mathcal{A})$ describes a distribution over actions for each state. Our goal is to learn the best policy $\pi^*$ that maximizes cumulative discounted reward, i.e., $\sum_t \mathbb{E}_{a_t \sim \pi^*} \gamma^t r(s_t, a_t)$. The value function and Quality(Q)-function of policy $\pi$ are $V^\pi(s) = \sum_t \mathbb{E}_{a_t \sim \pi(s_t)}[\gamma^t r(s_t, a_t)|s_0 = s]$, $Q^\pi(s, a) = \sum_t \mathbb{E}_{a_t \sim \pi(s_t)}[\gamma^t r(s_t, a_t)|s_0 = s, a_0 = a]$, respectively.

### 2.1. Offline-to-Online Reinforcement Learning

In O2O-RL, the agent undergoes two learning phases:

**Offline pre-training**: In the offline phase, the agent learns from a previously collected dataset $\mathcal{D}_{\text{offline}} = \{(s, a, r, s')\}$, possibly collected by another policy. Like recent offline RL methods (Levine et al., 2020), based on the recalibration of Q-value according to the self-consistency requirements dictated by the Bellman equation, i.e., $Q^\pi(s, a) = r(s, a) + \gamma \mathbb{E}_{s' \sim \mathbb{P}, a' \sim \pi}[Q^\pi(s', a')]$ (Watkins and Dayan, 1992), our work builds on approximate dynamic programming methods that minimize temporal difference error as follows:

$$\mathcal{L}_{TD}(\phi) := \mathbb{E}_{(s,a,s') \sim \mathcal{D}_{\text{offline}}} \left[ \left( r(s, a) + \gamma \max_{a'} Q_{\hat{\phi}}(s', a') - Q_\phi(s, a) \right)^2 \right], \quad (1)$$

where $Q_\phi(s, a)$ is a parameterized Q-function, $Q_{\hat{\phi}}(s, a)$ is a target network, and the policy is defined as $\pi(s) = \arg\max_a Q_\phi(s, a)$. In this offline phase, literature on O2O-RL often follows offline RL methods that modify the value function loss in Eq. 1 to regularize the value function so that the resulting policy is close to the behavior policy while maximizing the estimated Q-value (e.g., CQL (Kumar et al., 2020), IQL (Kostrikov et al., 2022), SO2 (Zhang et al., 2024), CalQL (Nakamoto et al., 2023), etc.). Notably, our method focuses on designing a diffusion actor policy and can combine with any of the modified critic methods above.

**Online fine-tuning**: In the online phase, the pre-trained agent continues to learn from interacting with the environments. Typically, for every time step $t$, the learned agent observes state $s_t$ and performs $a_t$ based on its policy $\pi(s_t)$. Then, the environment responds by transitioning to the next state $s_{t+1} \sim \mathbb{P}(s_{t+1}|s_t, a_t)$ and providing a re-

ward $r_t = r(s_t, a_t)$. The tuple $(s_t, a_t, r_t, s_{t+1})$ is stored in the online replay buffer $\mathcal{D}_{\text{online}}$, i.e., $\mathcal{D}_{\text{online}} = \mathcal{D}_{\text{online}} \cup (s_t, a_t, r_t, s_{t+1})$. After that, O2O-RL methods (Nakamoto et al., 2023; Liu et al., 2024; Huang et al., 2025) typically employ an actor-critic framework, i.e., the agent re-updates (fine-tunes) its knowledge by using a policy iteration process based on the experience replay buffer $\mathcal{D} = \mathcal{D}_{\text{offline}} \cup \mathcal{D}_{\text{online}}$. Specifically, in the policy evaluation phase, the Q-function $Q_\phi$ is optimized by the temporal difference error loss. Then, in the policy improvement phase, the policy function $\pi_\theta$ is sought by optimizing its parameter $\theta$ to maximize the corresponding current Q-value. Finally, this cycle repeats, with the agent using the uncertainty of policy $\pi_\theta$ balancing exploration (trying new actions) and exploitation (using known good actions) to maximize cumulative reward (return), i.e., $\sum_t \mathbb{E}_{a_t \sim \pi^*} \gamma^t r(s_t, a_t)$ in the online phase.

### 2.2. Diffusion planner in O2O-RL

Literature on Diffusion in O2O-RL (Liu et al., 2024; Huang et al., 2025) focuses on utilizing a diffusion planner (Janner et al., 2022) to augment trajectory data that conforms to the online distribution, thereby fine-tuning the above actor-critic model. Specifically, the diffusion planner represents trajectories as a single-channel image by:

$$\tau = \begin{bmatrix} \tau_{s_0} & \tau_{s_1} & \cdots & \tau_{s_{H-1}} & \tau_{s_H} \\ \tau_{a_0} & \tau_{a_1} & \cdots & \tau_{a_{H-1}} & \tau_{a_H} \end{bmatrix}, \quad (2)$$

where $\tau_{s_i}$ and $\tau_{a_i}$ are the state and action in the trajectory at time step $i \in [H]$, with $H$ being the trajectory length. Then, the diffusion planner tries to "diffuse" training data from the replay buffer $\mathcal{D}_{\text{offline}}$ with random noise $N$ times (forward), and learns to reverse the diffusion process to output new trajectories (backward) as follows:

**Forward diffusion** aims to "diffuse" training data with random noise. Specifically, given a training data $\tau^0 \sim q$, where $q$ is the probability distribution to be learned, then it repeatedly adds noise to it by $\tau^0 \to \tau^1 \to \cdots \to \tau^N$, by the Markov assumption, thus $q(\tau^{1:N}|\tau^0) = \prod_{n=1}^N q(\tau^n|\tau^{n-1})$, where $q(\tau^n|\tau^{n-1}) = \mathcal{N}(\tau^n; \sqrt{1-\beta_n}\tau^{n-1}, \beta_n \mathbf{I})$;

**Backward diffusion** tries to reverse the diffusion process to output new trajectories by $\tau^N \to \tau^{N-1} \to \cdots \to \tau^0$. The key idea of Denoising Diffusion Probabilistic Model (Ho et al., 2020) is to use a neural network $\epsilon_\theta$ parametrized by $\theta$ to map from $(\tau^n, n)$ to $\tau^{n-1} \sim \mathcal{N}(\mu_\theta(\tau^n, n), \Sigma_n)$, where $\mu_\theta(\tau^n, n) = \frac{\tau^n - \epsilon_\theta(\tau^n, n)\beta_n/\sqrt{1-\bar{\alpha}_n}}{\sqrt{\alpha_n}}$, $\bar{\alpha}_n = \alpha_1 \cdots \alpha_n$, and $\alpha_n = 1 - \beta_n$. This gives us a backward diffusion process $p_\theta$ defined by $p_\theta(\tau^{0:N}) := p(\tau^N) \prod_{n=1}^N p_\theta(\tau^{n-1}|\tau^n)$, where $p(\tau^N) = \mathcal{N}(\tau^N; 0, \mathbf{I})$.

**Objective function**: The goal now is to learn the parameters such that $p_\theta(\tau^0)$ is as close to $q(\tau^0)$ as possible. Unfortunately, $p_\theta(\tau^0) = \int p_\theta(\tau^{0:N}) d\tau^{1:N}$, where $\tau^{1:N}$ are latent variables, which is intractable. Hence, by using maximum

likelihood estimation with variational inference, the objective function can be derived from an Evidence Lower Bound (ELBO) and is optimized by:

$$\mathcal{L}_{\text{VLB}}(\theta, n) := \mathbb{E}_{\tau^0 \sim q, z \sim \mathcal{N}(0, \mathbf{I})} \left[ ||\epsilon_\theta(\tau^n, n) - z||^2 \right]. \quad (3)$$

After training the diffusion planner with Eq. 3 on $\mathcal{D}_{\text{offline}}$, Diffusion for O2O-RL, e.g., EDIS (Liu et al., 2024), use an energy guider $\varepsilon(\tau) = \frac{q(\tau^0)^{\text{online}}}{q(\tau^0)^{\text{offline}}}$ (i.e., a separated classifier that is trained from $\mathcal{D}_{\text{offline}} \cup \mathcal{D}_{\text{online}}$) to augment trajectory data that conforms to the online distribution by $p_\theta(\tau)^{\text{online}} \propto p_\theta(\tau)^{\text{offline}} e^{-\varepsilon(\tau)}$. Finally, the augmented data from $p_\theta(\tau)^{\text{online}}$ is added to the experience replay buffer to fine-tune another actor-critic model. Yet, this framework still suffers from two limitations, including the policy being a non-diffusion model and its inability to provide epistemic uncertainty. Hence, to address these issues, we introduce our new diffusion framework in the following section.

## 3. Efficient & Uncertainty-Aware Diffusion Actor-Critic O2O-RL Framework

In this section, we will thoroughly present two key components in our framework (summarized in Fig. 1 and Alg. 1), including an efficient diffusion actor policy from the diffusion planner (Sec. 3.1), and a novel method to quantify the uncertainty of this diffusion policy to balance exploration and exploitation in the online phase (Sec. 3.2 and Sec. 3.3).

### 3.1. Distillation and training a fast-sampling actor policy from diffusion planner

**Extract actor policy**: As introduced above, although the diffusion policy performs well in short-term planning, it often underperforms the diffusion planner in complex long-term planning RL tasks due to operating without lookahead planning (Lu et al., 2025; Chen et al., 2023). Therefore, motivated by the rich expression of diffusion planners for long-term planning RL tasks, we leverage the diffusion planner in O2O-RL (Liu et al., 2024; Huang et al., 2025) from Sec. 2.2 to extract our diffusion policy. Specifically, our policy extraction step is based on the following theorem:

**Theorem 3.1.** *(Lin et al., 2023) Consider the distribution of trajectory from the backward diffusion process $p_\theta(\tau)$ with a trajectory $\tau$ that is composed of a sequence of state-action pairs: $\tau = \{\tau_{s_0}, \tau_{a_0}, \tau_{s_1}, \tau_{a_1}, ..., \tau_{s_H}, \tau_{a_H}\}$. Then, the policy can be inferred from the distribution of trajectory $p_\theta(\tau)$ since the actions can be determined by*

$$\pi_\theta(a|s) \propto \int p_\theta(\tau_{>0}, \tau_{s_0} = s, \tau_{a_0} = a) d\tau_{>0},$$

*where $p_\theta(\tau_{>0}, \tau_{s_0} = s, \tau_{a_0} = a) \propto p_\theta(\tau_{>0}|\tau_{s_0} = s, \tau_{a_0} = a)$ and $\tau_{>0} := \{\tau_{s_1}, \tau_{a_1}, ..., \tau_{s_H}, \tau_{a_H}\}$. Proof is in Apd. A.1.*

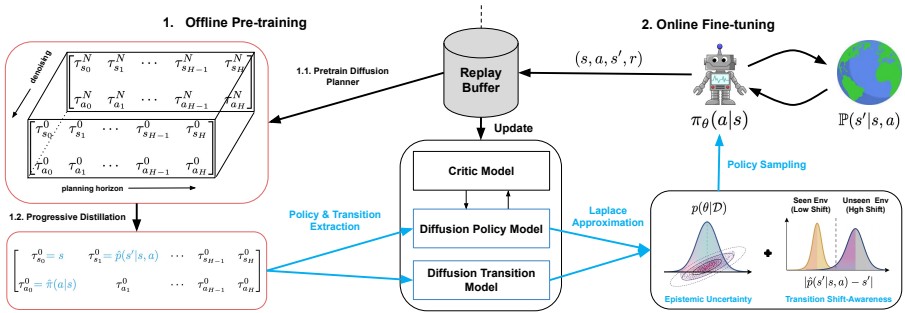

*Figure 1.* DUAL's overview framework, including an efficient diffusion actor policy & transition function extracted from diffusion planner, and a high-quality epistemic UQ & transition shift-awareness for policy sampling (colored by cyan). Overall algorithm is in Alg. 1.

Note that the result of Thm. 3.1 has also been used empirically in Lin et al. (2023). Thm. 3.1 suggests that the trajectory distribution from the diffusion planner model can be seen as a special diffusion policy. Specifically, the output action $a$ for a given input state $s$ can be extracted by adopting the first action of a trajectory sampled from the conditional distribution $p_\theta(\tau|\tau_{s_0} = s)$, i.e.,

$$a \sim \tau_{a_0}^0, \text{ where } \tau \sim p_\theta(\tau|\tau_{s_0} = s). \qquad (4)$$

**Progressive distillation for fast-sampling**: Training and inference with the diffusion policy above can incur huge computational costs when $N$ is high, and instability with low-quality trajectory outputs when $N$ is small, thus being expensive and not scalable in RL. To address this problem, we adapt the progressive distillation for diffusion training from Salimans and Ho (2022). In particular, given the teacher diffusion sampler $\epsilon_\theta(\tau^n; n)$ that maps random noise $\tau^N$ to samples $\tau^0$ in $N$ deterministic steps by $n \in \{N, N-1, N-2, \cdots, 3, 2, 1\}$, we will iteratively distill it into a new student sampler $\epsilon_{\hat\theta}(\tau^n; n)$ that maps random noise $\tau^N$ to samples $\tau^0$ in $N/2$ deterministic steps by $n \in \{N, N-2, \cdots, 3, 1\}$, until $N = 1$.

**Actor objective function**: After completing the policy extraction and progressive distillation, we are ready to present our policy actor objective function in training. Recall that in the offline phase, our goal is to train a policy close to the behavior policy while maximizing the estimated Q-value. First, it is worth noticing that the diffusion objective function in Eq. 3 results in a behavior-cloning objective for the policy $\pi_\theta$ by the following theorem:

**Theorem 3.2.** *Minimizing the diffusion planner's objective function in Eq. 3 on the offline buffer $\mathcal{D}_{offline} = \{(s, a, r, s')\}$ gives us an upper bound on the Kullback–Leibler (KL) Divergence of $\pi_{\mathcal{D}_{off}}(a|s)$ from $\pi_\theta(a|s)$ due to: (1) $D_{KL}\left(q(\tau^0) \| p_\theta(\tau^0)\right) \geq D_{KL}\left(\pi_{\mathcal{D}_{off}}(a|s) \| \pi_\theta(a|s)\right)$; (2) if $p_\theta$ is sufficiently expressive that $\min_\theta D_{KL}\left(q(\tau^0) \| p_\theta(\tau^0)\right) = 0$, then $\arg\min_\theta D_{KL}\left(q(\tau^0) \| p_\theta(\tau^0)\right) = \arg\min_\theta D_{KL}\left(\pi_{\mathcal{D}_{off}}(a|s) \| \pi_\theta(a|s)\right)$, where $\pi_{\mathcal{D}_{off}}(a|s)$ is the behavior policy in $\mathcal{D}_{offline}$. Proof is in Apd. A.2.*

Thm. 3.2 shows that the objective function in Eq. 3 helps the policy $\pi_\theta(a|s)$ close to the behavior policy (i.e., the policy that generated the static dataset) $\pi_{\mathcal{D}_{off}}(a|s)$ in offline RL. Hence, we only need to design an actor loss to maximize the estimated Q-value in the actor-critic framework in Sec. 2.1, thus we use the policy gradient with the loss defined by:

$$\mathcal{L}_\pi(\theta) = -\mathbb{E}_{(s,a)\sim\mathcal{D}}[Q_\phi(s, a) \cdot \log \pi_\theta(a|s)] - \log p(\theta), \qquad (5)$$

where $p(\theta)$ is a prior regularizer and is set to be uniformly distributed with density 1 in our experiment. And, $\pi_\theta(a|s) = \mathcal{N}(\tau_{a_0}^0, \sigma_{\bar\theta}^2)$ is parameterized as a Gaussian distribution, with the learnable mean $\tau_{a_0}^0$ obtained from Eq. 4 and $\sigma_{\bar\theta}^2$ is the corresponding learnable variance. Note that we can also extend to capture multimodal distribution by using a Gaussian Mixture Model for $\pi_\theta(a|s)$ in Eq. 5, though this will induce a higher computational training cost. From Eq. 5, we can see that it is a weighted maximum a-posteriori estimation (MAP) problem where samples are weighted by the critic $Q_\phi(s, a)$. This aims to optimize a policy by increasing the likelihood of actions leading to high rewards and decreasing it for actions that lead to low rewards. This objective is crucial and helps us design a proxy for a principled epistemic UQ for the policy $\pi_\theta$ in the following section.

### 3.2. Estimate Epistemic Uncertainty for Diffusion Policy

In online RL, the agent uses the policy's uncertainty to balance between trying new actions (exploration) to discover better rewards versus using known good actions (exploitation) for immediate gains (Lattimore and Szepesvári, 2020). Unfortunately, most policies are either deterministic or parametrized as a distribution with learned variance, only provide aleatoric (data) uncertainty (Chua et al., 2018; Bui et al., 2025), and are often overconfident under distribution shifts. Without epistemic (model) uncertainty, such overconfident models usually get stuck in suboptimal policies, thereby hindering exploration of the optimal policy in the online phase (Osband et al., 2023a;b; 2016).

Therefore, to obtain the epistemic uncertainty of $\pi_\theta(a|s)$, from the Bayesian perspective, we need to seek the posterior distribution $p(\theta|\mathcal{D})$ of the weight $\theta$ from the replay buffer

$\mathcal{D}$ (Kendall and Gal, 2017; Gal and Ghahramani, 2016; Bellemare et al., 2017). From Thm 3.1, since $\pi_\theta(a|s)$ is extracted from the distribution of diffusion trajectory $p_\theta(\tau)$, an approach may utilize existing diffusion UQ methods that employ the Laplace approximation with the ELBO loss in Eq. 3 (e.g., BayesDiff (Kou et al., 2024) or DiffUQ (Jazbec et al., 2025)) to the last layer of the denoising network $\epsilon_\theta$. Unfortunately, this is not a principled epistemic UQ by:

*Remark* 3.3. The Laplace approximation with the ELBO loss in Eq. 3, i.e., $p(\theta|\mathcal{D}) \approx \mathcal{N}(\hat\theta, \Sigma)$, where $\hat\theta = \arg\min_\theta \mathcal{L}_{\text{VLB}}(\theta)$ and $\Sigma = \left(\nabla_\theta^2 \mathcal{L}_{\text{VLB}}(\theta, n)|_{\theta=\hat\theta}\right)^{-1}$ does not align well with the standard Laplace's approximation framework, because the ELBO loss $\mathcal{L}_{\text{VLB}}$ may not be interpreted as a log-likelihood distribution, causing the posterior distribution $p(\theta|\mathcal{D}) \not\approx \frac{\exp(-\mathcal{L}_{\text{VLB}}(\theta))p(\theta)}{\int_{\theta'} \exp(-\mathcal{L}_{\text{VLB}}(\theta'))p(\theta')d\theta'}$.

From Rem. 3.3, we can also see that applying existing Diffusion UQ methods creates a misalignment in the objective, as $\mathcal{L}_{\text{VLB}}(\theta)$ does not explicitly account for downstream policy performance when approximating $p(\theta|\mathcal{D})$. This yields uninformative uncertainty estimates, degrading the UQ quality of the policy $\pi_\theta(a|s)$ needed for balancing exploration and exploitation. Therefore, to establish a principled epistemic UQ for policy $\pi_\theta$, such that the approximation objective aligns with the policy's decision-making process, we introduce our last-layer Laplace approximation as follows:

$$p(\theta|\mathcal{D}) \approx \mathcal{N}(\theta_{\text{MAP}}, \Sigma), \text{ where } \begin{cases} \theta_{\text{MAP}} = \arg\min_\theta \mathcal{L}_\pi(\theta), \\ \Sigma = \left(\nabla_\theta^2 \mathcal{L}_\pi(\theta)|_{\theta=\theta_{\text{MAP}}}\right)^{-1}. \end{cases} \tag{6}$$

Taking advantage of the weighted actor loss in Eq. 5, the approximation in Eq. 6 maintains strict consistency with a Bayesian interpretation of the epistemic uncertainty inherent in the policy. Indeed, we next show that our proposed approximation is more standard by the following theorem:

**Theorem 3.4.** *The Laplace approximation in Eq. 6 with the actor loss $\mathcal{L}_\pi(\theta)$ in Eq. 5 applies standard Laplace's approximation framework and results in a weighted posterior distribution $p(\theta|\mathcal{D}) \approx \frac{1}{Z} \exp(-\mathcal{L}_\pi(\theta))$ with the weighted marginal likelihood $Z = \exp(-\mathcal{L}_\pi(\theta_{MAP}))(2\pi)^{\frac{|\theta|}{2}}(\det(\Sigma))^{\frac{1}{2}}$. Proof is in Apd. A.3.*

*Note: We can derive the weighted posterior in Thm. 3.4 because $\mathcal{L}_\pi(\theta)$ in Eq. 5 is an off-policy loss, i.e., the posterior probability conditioned solely on the replay buffer $\mathcal{D}$ (Abdolmaleki et al., 2018). In our experiments, beyond off-policy loss in Eq. 5 with IQL (Kostrikov et al., 2022), we also extend our framework to other policy improvement variants, including SAC-style actor loss, i.e., $-\mathbb{E}_{s \sim \mathcal{D}, a \sim \pi}[Q_\phi(s,a) - \log \pi_\theta(a|s)]$ in Cal-QL (Nakamoto et al., 2023), and OFF2ON actor loss with on-policy samples in SO2+OFF2ON (Lee et al., 2022; Zhang et al., 2024).*

---

**Algorithm 1** DUAL in O2O-RL. Our proposed and optional steps are highlighted in cyan and gray, respectively.

---

1: Initialize the noise prediction of diffusion planner model $g_{\theta'}$, value functions $Q_\phi$, offline replay buffer $\mathcal{D}_{\text{offline}}$, online replay buffer $\mathcal{D}_{\text{online}}$.
2: **Offline phase:**
3: Pre-train diffusion planner $g_{\theta'}$ from $\mathcal{D}_{\text{offline}}$:
4: $\quad \theta' \leftarrow \theta' - \lambda_g \nabla_{\theta'} \mathcal{L}_{\text{VLB}}(\theta; \mathcal{D}_{\text{offline}})$
5: Distillate diffusion planner $g_{\theta'}$ to get $\pi_\theta, p(s'|s, a; \theta)$
6: Pre-train policy actor $\pi_\theta$ and critic $Q_\phi$ from $\mathcal{D}_{\text{offline}}$:
7: $\quad \phi \leftarrow \phi - \lambda_Q \nabla_\phi \mathcal{L}^Q_{\mathcal{D}_{\text{offline}}}(\phi)$
8: $\quad \theta \leftarrow \theta - \lambda_\pi \nabla_\theta \mathcal{L}^\pi_{\mathcal{D}_{\text{offline}}}(\theta)$
9: Run Laplace approximation for $\pi_\theta$ from $\mathcal{L}^\pi_{\mathcal{D}_{\text{offline}}}(\theta)$
10: **Online phase:**
11: **while** in *online training phase* **do**
12: $\quad$ Sample $a_t \sim \frac{1}{M} \sum_{m=1}^M \pi_{\theta_m}(\cdot|s_t) + \lambda \mathcal{N}(0, ||s_t - p_\theta(\hat{s}_t|s_{t-1}, a_{t-1})||)$
13: $\quad$ Receive reward $r_t$ and new state $s_{t+1}$
14: $\quad \mathcal{D}_{\text{online}} \leftarrow \mathcal{D}_{\text{online}} \cup \{(s_t, a_t, s_{t+1}, r_t)\}$
15: $\quad$ **if** step meets update frequency **then**
16: $\quad\quad$ Add generated samples from $g_\theta$ to $\mathcal{D}_{\text{online}}$ (e.g., EDIS)
17: $\quad\quad$ **for** each gradient step **do**
18: $\quad\quad\quad$ Sample data from $\mathcal{D} = \mathcal{D}_{\text{online}} \cup \mathcal{D}_{\text{offline}}$
19: $\quad\quad\quad \phi \leftarrow \phi - \lambda_Q \nabla_\phi \mathcal{L}^Q_{\mathcal{D}}(\phi)$
20: $\quad\quad\quad \theta \leftarrow \theta - \lambda_\pi \nabla_\theta \mathcal{L}^\pi_{\mathcal{D}}(\theta)$
21: $\quad\quad$ **end for**
22: $\quad\quad$ Run Laplace approximation $\pi_\theta$ from $\mathcal{L}^\pi_{\mathcal{D}}(\theta)$
23: $\quad$ **end if**
24: **end while**

---

Thm. 3.4 confirms our proposed approximation in Eq. 6 can provide accurate weighted posterior distribution $p(\theta|\mathcal{D})$ for the weight $\theta$. We next show how to use this principled posterior for policy sampling with epistemic uncertainty to balance exploration and exploitation in online RL.

### 3.3. Policy sampling: Epistemic UQ & Shift-awareness

**Epistemic UQ**: Given the weighted posterior distribution $p(\theta|\mathcal{D})$, the epistemic uncertainty of policy $\pi_\theta(a|s)$ can then be defined as the variability of the posterior predictive: $\mathcal{V}(\pi_\theta(a|s))$, where $\mathcal{V}(\cdot)$ denotes the variability measure, such as variance, entropy, etc. Recall that from Thm 3.1, the policy can be inferred from the diffusion planner by $\pi_\theta(a|s) \propto \int p_\theta(\tau, \tau_{s_0} = s, \tau_{a_0} = a)d\tau$, hence, to obtain $\mathcal{V}(\pi_\theta(a|s))$ from $p(\theta|\mathcal{D})$, we can apply Monte-Carlo sampling to maintain $M$ sub-policies $\{\pi_{\theta_1}, \pi_{\theta_2}, \ldots, \pi_{\theta_M}\}$, where $\pi_{\theta_m}(a|s) \propto \int p_{\theta_m}(\tau, \tau_{s_0} = s, \tau_{a_0} = a)d\tau$, $\theta_m \sim p(\theta|D)$ for every $m \in [M]$. Then, the final ensemble policy $\hat\pi$ can be derived through mean and variance aggregation over the sub-policies as follows:

$$\hat\pi_\theta(\cdot|s) = \frac{1}{M} \sum_{m=1}^M \pi_{\theta_m}(\cdot|s) = \mathcal{N}(\mu_\theta(s), \sigma_\theta^2(s)),$$

$$\text{where } \begin{cases} \mu_\theta(s) = \frac{1}{M} \sum_{m=1}^M (\tau_{a_0}^0)_m, \\ \sigma_\theta^2(s) = \frac{1}{M} \sum_{m=1}^M \left[\left(\mu_\theta(s) - (\tau_{a_0}^0)_m\right)^2 + \sigma_{\theta_m}^2\right], \end{cases} \tag{7}$$

where $(\tau)_m \sim p_{\theta_m}(\tau|\tau_{s_0} = s)$ for every $m \in [M]$. With the ensemble policy $\hat{\pi}$ in Eq. 7, when interacting with the environment, the agent can simply sample the action $a \sim \hat{\pi}_\theta(\cdot|s)$. Therefore, we can guarantee the action $a$ is sampled by the epistemic uncertainty of policy $\pi_\theta(a|s)$ by the variance measure $\mathcal{V}(\pi_\theta(a|s)) = \sigma_\theta^2(s)$ in Eq. 7.

**Shift-awareness:** While literature on O2O-RL focuses on the distribution shift caused by the difference of offline and online policy (Wang et al., 2023a), it can be the case that the transition-state distribution $\mathbb{P}(s'|s, a)$ is different (Jin et al., 2018; Lyu et al., 2024; Bui et al., 2026). E.g., in robot navigation on Frozen-Lake, an offline dataset can be collected from a certain slippery level in $\mathbb{P}(s'|s, a)$. Yet, due to weather conditions, the slippery levels can subsequently change in the online phase. Therefore, to further enhance the policy's UQ quality under the transition shift, we propose an additional shift-awareness term for policy sampling by:

$$\hat{\pi}_\theta(\cdot|s) = \mathcal{N}\left(\mu_\theta(s), \sigma_\theta^2(s) + \lambda|\hat{p}(s'|s, a) - s'|\right), \quad (8)$$

where $\lambda$ is the weight hyper-parameter (ablation study with $\lambda$ is in Apd. B.2), and $\hat{p}(s'|s, a)$ is the estimated transition model. It is worth noticing that, based on the derivation in the proof of Thm. 3.1, we can extract $\hat{p}(s'|s, a)$ from diffusion planner for free by $p(s'|s, a) \propto \int p_\theta(\tau, \tau_{s_0} = s, \tau_{a_0} = a)d\tau$. From Eq. 8, we can see that the second term of variance, i.e., $|\hat{p}(s'|s, a) - s'|$, measures the difference between the estimated state and the true state. Therefore, when this value is high, $\hat{\pi}_\theta(s)$ will be more uncertain (exploration), and when there is no transition shift, this value is small, yielding $\hat{\pi}_\theta(\cdot|s)$ more certain (exploitation). As a result, policy sampling in Eq. 8 can provide both epistemic

UQ and shift-awareness ability to balance exploration and exploitation in O2O-RL.

## 4. Experiments

### 4.1. Benchmark comparison

We compare our method (i.e., DUAL) with diffusion-based RL baselines in the O2O-RL setting. Specifically, EDIS (Liu et al., 2024) uses diffusion as a data-synthesizer with an energy guider for actor-critic in online-finetuning; Diffuser (Janner et al., 2022) uses a diffusion model for planning with a separate return guider; Diff-QL (Wang et al., 2023b) utilizes a conditional diffusion to represent the actor policy in the actor-critic framework; DACER (Wang et al., 2024) is an extension of Diff-QL by adding an entropy regularizer to quantify UQ to balance exploration exploitation in the online phase. Details are in Apd. B.1.

Tab. 1 shows the average expected online return on MuJoCo locomotion and AntMaze environments with D4RL dataset. First, we observe that using a diffusion policy within the actor-critic framework can yield significant improvements. For example, in MuJoCo, where the goal is to control agents (e.g., humanoids, quadrupeds) to stand stably or run faster, diffusion policy methods such as Diff-QL and DACER achieve returns exceeding 86. Notably, our method leverages the diffusion policy with a high-quality epistemic UQ, which can balance the exploration and exploitation, resulting in an improvement of more than 89.9 in expected return in the online phase. Second, by extracting the policy from the prior knowledge of the diffusion planner, our policy outperforms significantly other methods by more than

*Table 1.* Average online expected returns on MuJoCo and AntMaze environments after $1M$ time steps from Fig. 4 and Fig. 5. Each result is the average score over five random seeds with $\pm$ standard deviation. Best scores with the significant test ($\alpha = 0.05$) are marked in **bold**.

| Dataset | IQL | EDIS | Diffuser | Diff-QL | DACER | DUAL (Ours) |
|---|---|---|---|---|---|---|
| halfcheetah-medium-v2 | $48.35 \pm 1.19$ | $49.34 \pm 1.39$ | $48.20 \pm 0.92$ | $48.61 \pm 1.37$ | $49.92 \pm 0.89$ | $49.83 \pm 1.09$ |
| halfcheetah-medium-replay-v2 | $45.77 \pm 1.24$ | $46.65 \pm 1.38$ | $45.82 \pm 1.87$ | $46.91 \pm 1.70$ | $\mathbf{48.18} \pm 1.00$ | $\mathbf{48.78} \pm 1.30$ |
| halfcheetah-medium-expert-v2 | $95.77 \pm 1.52$ | $98.24 \pm 1.91$ | $96.60 \pm 1.91$ | $99.88 \pm 1.79$ | $\mathbf{102.18} \pm 1.37$ | $\mathbf{103.11} \pm 1.75$ |
| hopper-medium-v2 | $66.61 \pm 2.68$ | $68.90 \pm 2.71$ | $70.43 \pm 2.27$ | $80.26 \pm 2.69$ | $83.05 \pm 2.29$ | $\mathbf{88.23} \pm 3.42$ |
| hopper-medium-replay-v2 | $101.01 \pm 2.13$ | $103.55 \pm 2.46$ | $96.57 \pm 3.55$ | $\mathbf{108.27} \pm 2.42$ | $\mathbf{110.06} \pm 2.01$ | $\mathbf{109.21} \pm 2.41$ |
| hopper-medium-expert-v2 | $107.48 \pm 3.29$ | $109.93 \pm 3.41$ | $106.73 \pm 3.33$ | $110.94 \pm 4.16$ | $\mathbf{113.88} \pm 3.57$ | $\mathbf{115.33} \pm 4.58$ |
| walker2d-medium-v2 | $84.20 \pm 1.61$ | $86.11 \pm 1.81$ | $83.57 \pm 1.77$ | $86.14 \pm 1.74$ | $\mathbf{90.05} \pm 1.39$ | $\mathbf{89.15} \pm 1.64$ |
| walker2d-medium-replay-v2 | $84.04 \pm 1.67$ | $89.53 \pm 1.85$ | $85.51 \pm 2.13$ | $87.88 \pm 1.80$ | $90.83 \pm 1.44$ | $\mathbf{93.15} \pm 1.86$ |
| walker2d-medium-expert-v2 | $111.52 \pm 1.28$ | $111.02 \pm 1.31$ | $111.87 \pm 0.83$ | $\mathbf{112.48} \pm 0.92$ | $\mathbf{112.43} \pm 0.96$ | $\mathbf{112.62} \pm 1.18$ |
| **locomotion average** | 82.7500 | 84.8078 | 82.8111 | 86.8189 | 88.9533 | **89.9344** |
| antmaze-umaze-v2 | $90.38 \pm 2.24$ | $90.10 \pm 1.20$ | $89.75 \pm 1.24$ | $90.91 \pm 1.45$ | $91.31 \pm 1.82$ | $\mathbf{94.69} \pm 4.00$ |
| antmaze-umaze-diverse-v2 | $35.95 \pm 15.16$ | $35.82 \pm 11.62$ | $35.71 \pm 10.36$ | $49.37 \pm 10.78$ | $51.73 \pm 7.94$ | $\mathbf{78.06} \pm 9.03$ |
| antmaze-medium-play-v2 | $85.03 \pm 2.63$ | $84.15 \pm 2.00$ | $86.45 \pm 1.70$ | $87.44 \pm 1.77$ | $87.46 \pm 1.18$ | $\mathbf{92.29} \pm 2.40$ |
| antmaze-medium-diverse-v2 | $84.10 \pm 2.25$ | $83.55 \pm 2.80$ | $86.58 \pm 1.84$ | $86.75 \pm 1.52$ | $86.93 \pm 1.68$ | $\mathbf{91.86} \pm 2.76$ |
| antmaze-large-play-v2 | $54.16 \pm 4.00$ | $56.80 \pm 3.70$ | $59.21 \pm 2.32$ | $58.40 \pm 3.65$ | $60.50 \pm 3.25$ | $\mathbf{70.27} \pm 4.72$ |
| antmaze-large-diverse-v2 | $50.21 \pm 3.83$ | $53.57 \pm 5.02$ | $55.64 \pm 3.05$ | $51.13 \pm 3.55$ | $52.30 \pm 4.47$ | $\mathbf{64.21} \pm 4.63$ |
| **antmaze average** | 66.6383 | 67.3317 | 68.8900 | 70.6667 | 71.7050 | **81.8967** |

*Table 2.* Average online expected return on MuJoCo and off-dynamic (ODRL) environment after $1M$ time steps from Fig. 5. Each result is the average score over five random seeds with $\pm$ standard deviation. Best scores with the significant test ($\alpha = 0.05$) are marked in **bold**.

| Dataset | Cal-QL | | | | SO2+OFF2ON | | | |
|---|---|---|---|---|---|---|---|---|
| | Base | EDIS | Diff-QL | DUAL (Ours) | Base | Diff-QL | DACER | DUAL (Ours) |
| halfcheetah-medium-v2 | $74.14 \pm 1.39$ | $80.18 \pm 1.55$ | $80.37 \pm 1.28$ | $\mathbf{84.61} \pm 1.86$ | $77.72 \pm 1.01$ | $79.48 \pm 1.43$ | $\mathbf{80.00} \pm 0.77$ | $81.78 \pm 1.11$ |
| halfcheetah-medium-replay-v2 | $75.20 \pm 0.95$ | $82.16 \pm 1.28$ | $84.74 \pm 1.21$ | $\mathbf{86.99} \pm 2.00$ | $68.70 \pm 0.85$ | $70.18 \pm 1.25$ | $72.50 \pm 0.77$ | $\mathbf{74.42} \pm 1.20$ |
| hopper-medium-v2 | $98.21 \pm 1.93$ | $104.51 \pm 2.30$ | $102.04 \pm 1.33$ | $\mathbf{107.95} \pm 1.77$ | $88.84 \pm 2.33$ | $94.30 \pm 1.73$ | $96.97 \pm 1.53$ | $\mathbf{100.78} \pm 1.59$ |
| walker2d-medium-v2 | $98.77 \pm 1.81$ | $103.54 \pm 2.06$ | $106.80 \pm 1.53$ | $\mathbf{109.10} \pm 1.93$ | $88.87 \pm 0.53$ | $91.02 \pm 1.37$ | $92.38 \pm 0.46$ | $\mathbf{94.45} \pm 0.58$ |
| **D4RL locomotion average** | 86.5800 | 92.5975 | 93.4875 | **97.1625** | 81.0325 | 83.7450 | 85.4600 | **87.8575** |
| halfcheetah-friction-2.0 | $61.20 \pm 1.19$ | $62.65 \pm 0.51$ | $62.68 \pm 1.31$ | $\mathbf{67.16} \pm 1.02$ | $56.60 \pm 0.90$ | $57.35 \pm 1.25$ | $60.21 \pm 1.29$ | $\mathbf{61.94} \pm 0.62$ |
| halfcheetah-med-gravity-2.0 | $44.98 \pm 1.08$ | $46.19 \pm 0.65$ | $47.48 \pm 1.63$ | $\mathbf{53.41} \pm 1.10$ | $42.52 \pm 2.61$ | $45.02 \pm 2.66$ | $47.93 \pm 2.72$ | $\mathbf{49.97} \pm 1.49$ |
| hopper-kinematic-2.0 | $55.46 \pm 3.01$ | $58.01 \pm 2.44$ | $62.93 \pm 2.09$ | $\mathbf{68.70} \pm 1.86$ | $51.63 \pm 5.95$ | $59.15 \pm 3.43$ | $61.55 \pm 3.64$ | $\mathbf{63.00} \pm 2.20$ |
| walker2d-morphology-2.0 | $52.70 \pm 1.70$ | $55.17 \pm 0.59$ | $56.93 \pm 2.37$ | $\mathbf{61.22} \pm 0.92$ | $48.96 \pm 4.01$ | $53.01 \pm 4.03$ | $56.23 \pm 4.25$ | $\mathbf{58.63} \pm 2.51$ |
| **ODRL locomotion average** | 53.5850 | 55.5050 | 57.5050 | **62.6225** | 49.9275 | 53.6325 | 56.4800 | **58.3850** |

*Table 3.* Average online expected return on Adroit environment after $1M$ time steps from Fig. 5. Each result is the average score over five random seeds with $\pm$ standard deviation. Best scores with the significant test ($\alpha = 0.05$) are marked in **bold**.

| Dataset | IQL | | | | Cal-QL | | | |
|---|---|---|---|---|---|---|---|---|
| | Base | EDIS | Diff-QL | DUAL (Ours) | Base | EDIS | Diff-QL | DUAL (Ours) |
| pen-cloned-v1 | $94.74 \pm 12.12$ | $97.31 \pm 12.73$ | $104.45 \pm 12.14$ | $\mathbf{124.40} \pm 19.09$ | $-2.69 \pm 0.14$ | $-2.23 \pm 0.30$ | $-1.93 \pm 0.47$ | $\mathbf{-1.09} \pm 0.35$ |
| door-cloned-v1 | $9.79 \pm 4.98$ | $10.46 \pm 5.10$ | $11.55 \pm 4.93$ | $\mathbf{15.26} \pm 4.86$ | $-0.33 \pm 0.01$ | $-0.32 \pm 0.01$ | $-0.31 \pm 0.01$ | $\mathbf{-0.28} \pm 0.01$ |
| hammer-cloned-v1 | $27.41 \pm 13.36$ | $28.00 \pm 13.42$ | $33.82 \pm 12.10$ | $\mathbf{46.74} \pm 9.85$ | $0.26 \pm 0.28$ | $0.35 \pm 0.31$ | $0.32 \pm 0.21$ | $\mathbf{0.68} \pm 0.23$ |
| relocate-cloned-v1 | $0.10 \pm 0.05$ | $0.14 \pm 0.06$ | $0.23 \pm 0.05$ | $\mathbf{0.44} \pm 0.06$ | $-0.28 \pm 0.08$ | $-0.24 \pm 0.08$ | $-0.26 \pm 0.02$ | $\mathbf{-0.12} \pm 0.08$ |
| **adroit average** | 33.0100 | 33.9775 | 37.5125 | **46.7100** | -0.7600 | -0.6100 | -0.5450 | **-0.2025** |

10 return in long-term planning tasks like AntMaze, where the task's goal is to train a four-legged ant agent to navigate across different 2D mazes to reach a target location.

Next, we combine DUAL with two other base models with different critic Q-functions, including Cal-QL (Nakamoto et al., 2023) and SO2+OFF2ON (Zhang et al., 2024). Tab. 2 summarizes the results with MuJoCo under two settings, including D4RL and off-dynamic RL (i.e., ODRL). Overall, we observe a consistent and significant improvement of DUAL across different tasks, especially in off-dynamic settings, where the transition state (e.g., friction, gravity, kinematic, and morphology) distributions significantly change the robot motion characteristics between the offline and online phases. It is worth noticing that while Cal-QL uses SAC-style actor loss, SO2+OFF2ON uses near OFF2ON actor loss with on-policy samples (Lee et al., 2022). Hence, the result in Tab. 2 also suggests that beyond off-policy improvement loss, our framework can also empirically extend and bring out benefits for other on-policy variants.

In Tab. 3, we continue to extend our experiments to different datasets with the Adroit task. Similar to AntMaze, Adroit is an environment that is more closely related to the lookahead planning and requires intricate planning (e.g., opening a door with a latch, using a hammer to hit a nail, repositioning a pen in the hand, or moving a ball to a target position). Hence, we observe that our method can bring out a significant improvement over other diffusion-based methods. It is also worth noticing that this result is consistent in both IQL

and Cal-QL critic baselines.

### 4.2. Computational efficiency

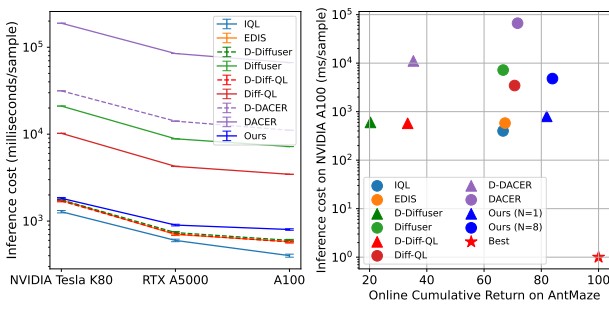

*Figure 2.* (a) Latency of policy sampling across different GPUs; (b) 2-D visualizations regarding online expected return (x-axis) & inference cost (y-axis).

Fast inference is a necessary condition for an efficient diffusion model, and is crucial in real-time online RL. Fig. 2 (a) compares the inference cost with different denoising time steps. We observe that our method, with a denoising timestep $N = 1$ and planning horizon $H = 8$, is much faster than other diffusion-based baselines in the benchmarking comparison section. This is because Diffuser and Diff-QL require $N = 20$ and $N = 8$ denoising time steps, respectively. In addition, our method is only slightly slower than their progressive distillation version with $N = 1$ (i.e., D-Diffuser and D-Diff-QL) by requiring MC sampling to

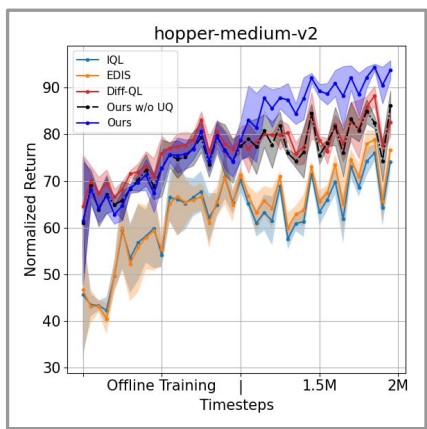 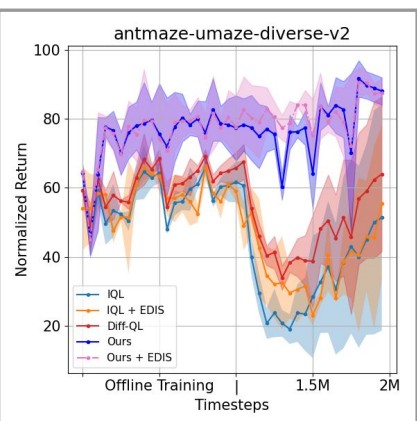 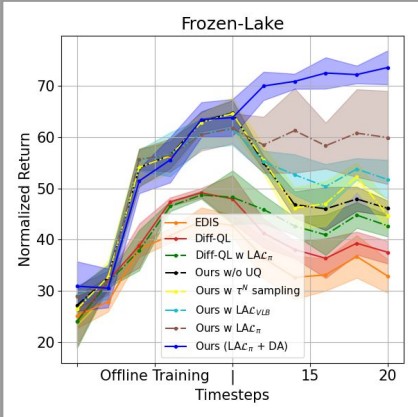

*Figure 3.* Normalized return in test set across $1M$ timesteps in offline training and $1M$ online finetuning phase in (a) MuJoCo Locomotion (hopper-medium-v2); (b) AntMaze (antmaze-umaze-diverse-v2); (c) Frozen-Lake (slipper level change between offline and online phase).

get epistemic UQ. Although D-Diffuser and D-Diff-QL are computationally efficient, their expected return degrades considerably in Fig. 2 (b). Regarding DACER, it is much slower than other baselines, even with the distillation version, i.e., D-DACER. This is because D-DACER requires hundreds of action samples for UQ with an entropy regularizer. Finally, we compare the gap between computational efficiency and expected return in our method. We observe that when the denoising steps increase from $N = 1$ to $N = 8$, the expected return improves from around $81.8$ to $84.0$. Yet, the latency also increases by around $6$ times. Although our method also suffers from the denoising time step trade-off, we observe that the efficiency and return gap are still smaller than other diffusion-based baselines in Fig. 2 (b).

### 4.3. Ablation studies

*Table 4.* Testing causal effect of each component in DUAL, including (i) planner-prior extraction itself, (ii) the distilled/efficient actor parameterization, (iii) uncertainty-aware sampling, and (iv) shift-aware adaptation. Evaluations on D4RL hopper-medium and ODRL hopper-friction environments.

| Method | hopper-medium | hopper-friction | Latency (ms/s) |
|---|---|---|---|
| w/o planner | $82.08 \pm 3.81$ | $35.41 \pm 3.44$ | $777.22 \pm 16$ |
| w/o distillation | $92.31 \pm 3.26$ | $53.26 \pm 3.30$ | $24953.11 \pm 30$ |
| w/o epistemic UQ | $83.36 \pm 3.85$ | $30.79 \pm 3.91$ | $775.18 \pm 16$ |
| w/o shift-aware | $87.30 \pm 3.44$ | $35.12 \pm 3.80$ | $798.93 \pm 16$ |
| DUAL (Ours) | $88.23 \pm 3.42$ | $48.00 \pm 3.44$ | $800.22 \pm 16$ |

**Causal effect of each component in DUAL**. To understand our method, we test the causal effect of each component in the Tab. 4. First, we observe that, without epistemic UQ sampling, the performance drops significantly. This shows that the epistemic UQ is the main contributor to our overall framework. Similarly, the planner-prior extraction also contributes significantly to the overall performance. Second, although the shift-aware adaptation contributes slightly to

the D4RL hopper-medium, it is an important component in the ODRL transition-shift setting (e.g., hopper-medium-friction). Finally, although the no-distillation variant yields slightly higher online return, using the distillation component is crucial in terms of reducing inference cost. Overall, this ablation study suggests that epistemic UQ sampling (i.e., (iii)) is the most important component. The planner-prior horizon (i.e., (i)) is the second important component, especially in longer planning tasks. The shift-awareness (i.e., (iv)) is crucial when transition shifts. The distillation (i.e., (ii)) is crucial in real-time applications.

**Performance across offline and online phases**. Next, we plot the evaluated return across O2O-RL time steps in Fig. 3. Regarding offline pre-training, we observe that on long-term planning tasks (e.g., AntMaze, Frozen-Lake), our method also outperforms other diffusion baselines. This could be because we extract our policy from the diffusion planner, so the policy can learn the prior knowledge about longer planning rewards. For tasks less reliant on lookahead planning (e.g., MuJoCo), our method without the UQ version performs close to the Diff-QL baseline in the offline phase and is also similar in the online phase. That said, with the UQ, our method outperforms significantly other baselines in the online phase, confirming the importance of our UQ to enhance O2O-RL performance.

**Our model extension to O2O-RL data-augmentation**. Fig. 3 (b) shows that our method can effectively combine with EDIS to improve the online return. In particular, after the online time step meets update frequency, we can use the pre-trained diffusion planner to generate data close to the online distribution in EDIS (Liu et al., 2024) to augment the replay buffer (i.e., the optional step in Alg. 1). Notably, while this additional step is more costly, it does not change our framework, and DUAL is compatible with this data augmentation method to boost O2O-RL performance.

**Contribution of UQ components in online phase**. We examine several UQ variants on top of our framework. First, we compare with a non-model UQ version which obtains sub-policies from sampling the input latent $\tau^N \sim \mathcal{N}(0, 1)$, then aggregating over these sub-policies in our Eq. 7. From Fig. 3 (c), we observe that this version (i.e., "Ours w $\tau^N$ sampling") can not improve the performance in online RL, and is close to the version without UQ. Fig. 3 (c) also confirms Thm. 3.4 that our principled epistemic UQ with actor loss in Eq. 6 (i.e., "Ours w LA$\mathcal{L}_\pi$") can result in better exploration and exploitation than the non-principled ELBO-based Laplace in Rmk. 3.3 (i.e., "Ours w LA$\mathcal{L}_{VLB}$"). Furthermore, in this Frozen-Lake, since we change the slippery levels of the environment, we can see that our shift-aware UQ component (i.e., "Ours w LA$\mathcal{L}_\pi$ + DA") can help the policy focus more on exploration when the transition-state shift happens, yielding a higher online return than the one without shift-awareness (i.e., "Ours w LA$\mathcal{L}_\pi$"). Finally, we apply our Laplace approximation with actor loss in Eq. 6 for Diff-QL, and observe that this "Diff-QL w LA$\mathcal{L}_\pi$" also has a higher online return than Diff-QL, showing the generality of our UQ contribution for diffusion-policy models.

## 5. Related work

**Offline-to-online Reinforcement Learning**. To mitigate the shift between offline and online distribution in O2O-RL, prior works often leverage both offline data and online data in replay buffer for online fine-tuning, and combine with methods in the process of Q-learning (Kumar et al., 2020; Nakamoto et al., 2023; Lee et al., 2022), policy improvement, policy expansion (i.e., freezes the pre-trained policy and initializes a random policy to enhance exploration) (Zhang et al., 2023), adaptive weight in online fine-tuning (Wang et al., 2023a), and other actor-critics alignment (Yu and Zhang, 2023; Wu et al., 2022; Fujimoto et al., 2019; Fujimoto and Gu, 2021; Kostrikov et al., 2022). Empirically, IQL (Kostrikov et al., 2022), Cal-QL (Nakamoto et al., 2023), and SO2 (Zhang et al., 2024) have become standard O2O-RL baselines to balance between being optimistic about improving the policy during the online phase and still being constrained to the conservative offline policy (Tarasov et al., 2022; Liu et al., 2024). In particular, IQL (Kostrikov et al., 2022) warms up the actor-critic by incorporating a weighted behavioral cloning step to enhance online policy improvement. Cal-QL (Nakamoto et al., 2023) value function calibration learns conservative value functions that are constrained to be larger than the value function of a reference policy to facilitate fast online fine-tuning. SO2 (Zhang et al., 2024) perturbs the update of the value to smooth out the estimation of biased Q-value and increases the frequency of updates to prevent the exploitation of actions in the early stages of the policy, and alleviates the estimation bias inherited from offline pretraining.

**Diffusion Model in Reinforcement Learning**. Using the diffusion model as either a planner, policy, or data synthesizer has shown promising results in offline RL due to the rich expressiveness of the diffusion model from the abundance of offline datasets (Zhu et al., 2024b; Lu et al., 2025). Starting with the diffusion planner, Diffuser (Janner et al., 2022; Ajay et al., 2023) uses the diffusion model to sample the whole trajectory with state and action pairs with useful properties for long-term planning tasks. Later on, Diff-QL (Wang et al., 2023b) uses the diffusion model as the parametrized policy class to output single-step actions, where the sampling target is the action conditioned on the state, usually guided by the Q-function via policy gradient-style guidance. Regarding online RL, DACER (Wang et al., 2024) has recently extended Diff-QL with an Entropy Regulator to balance exploration and exploitation. Although outperforming diffusion planner methods in short-term planning, diffusion policy approaches often underperform in long-term planning tasks (Lu et al., 2025). In O2O-RL, Liu et al. (2024); Huang et al. (2025) train a diffusion planner on an offline dataset to augment trajectory data that conforms to the online distribution, thereby fine-tuning the actor-critic model. Yet, the actor-critic is non-diffusion, omitting the rich expressiveness of the diffusion planner. To address this issue, we introduce DUAL, a unified diffusion framework for O2O-RL, resulting in an improvement in both short-term and long-term planning tasks.

## 6. Conclusion

O2O-RL has emerged as a promising approach to minimizing the costly interactions of Deep-RL models in real-world settings. Existing work still omits the rich expressiveness of the diffusion planner, and the pre-trained policy provides only aleatoric uncertainty, leading to suboptimal exploitation in the online phase. We introduce DUAL, an efficient Diffusion Uncertainty-Aware framework for O2O-RL. DUAL utilizes the prior knowledge of the diffusion planner to distill a fast-sampling diffusion actor policy and transition model in the offline phase. Based on these models, DUAL employs a novel epistemic uncertainty-aware technique that leverages the actor Laplace approximation and distance transition-state-shift detection, thereby enhancing exploration and exploitation performance in the online phase. Yet, our work still has limitations, including additional training costs, the gap between computational efficiency and expected return, the Gaussian assumption underlying the extracted diffusion policy, and additional latency in the online phase due to MC policy sampling. That said, with our promising results, we hope our work will contribute to the literature on improving the uncertainty and robustness of the diffusion model in sequential decision problems. Future work includes addressing the aforementioned limitations and exploring Diffusion Language Models for RL.

## Acknowledgment

This work is partially supported by an Amazon Research Award and a grant from JHU Institute of Assured Autonomy. We thank anonymous reviewers for their valuable feedback.

## Impact Statement

Our work tackles the quality and computational trade-off of diffusion RL models. It also provides a high-quality epistemic UQ, helping to balance the exploration-exploitation dilemma in RL. The broader impact includes advancing diffusion RL models in sequential decision-making, e.g., autonomous systems, robotics, finance, healthcare, and other high-stakes forecasting domains.

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

## A. Proofs

### A.1. Proof of Theorem 3.1

*Proof.* Consider three random variables, including the action $a$, state $s$, and trajectory $\tau$, by the definition of conditional probability, we have

$$\pi_\theta(a|s) = \frac{p_\theta(a,s)}{p_\theta(s)} = \frac{\int p_\theta(a,s,\tau)d\tau}{p_\theta(s)}. \tag{9}$$

On the other hand, by the definition of the trajectory in the diffusion planner, we have $\tau = \{\tau_{s_0}, \tau_{a_0}, \tau_{s_1}, \tau_{a_1}, ..., \tau_{s_H}, \tau_{a_H}\}$, since we define $\tau_{s_0} = s$ and $\tau_{a_0} = a$, yielding

$$p_\theta(a,s,\tau) = p_\theta(\tau_{>0}, \tau_{s_0} = s, \tau_{a_0} = a) \text{ and } p_\theta(\tau_{>0}|\tau_{s_0} = s, \tau_{a_0} = a) = p_\theta(\{\tau_{s_1}, \tau_{a_1}, ..., \tau_{s_H}, \tau_{a_H}\}|\tau_{s_0} = s, \tau_{a_0} = a). \tag{10}$$

Hence, we get

$$\pi_\theta(a|s) = \frac{\int p_\theta(\tau_{>0}, \tau_{s_0} = s, \tau_{a_0} = a)d\tau_{>0}}{p_\theta(s)} = \frac{\int p_\theta(\tau_{>0}|\tau_{s_0} = s, \tau_{a_0} = a)p_\theta(\tau_{s_0} = s, \tau_{a_0} = a)d\tau_{>0}}{p_\theta(s)} \tag{11}$$

$$= \frac{\int p_\theta(\{\tau_{s_1}, \tau_{a_1}, ..., \tau_{s_H}, \tau_{a_H}\}|\tau_{s_0} = s, \tau_{a_0} = a)p_\theta(\tau_{s_0} = s, \tau_{a_0} = a)d\tau_{>0}}{p_\theta(s)}, \tag{12}$$

by $p_\theta(s)$ is a constant w.r.t. $a$ and $p_\theta(\tau_{s_0} = s, \tau_{a_0} = a)$ is a constant w.r.t. $\{\tau_{s_1}, \tau_{a_1}, ..., \tau_{s_H}, \tau_{a_H}\}$, we obtain the result

$$\pi_\theta(a|s) \propto \int p_\theta(\tau_{>0}, \tau_{s_0} = s, \tau_{a_0} = a)d\tau_{>0}, \tag{13}$$

where $p_\theta(\tau_{>0}, \tau_{s_0} = s, \tau_{a_0} = a) \propto p_\theta(\tau_{>0}|\tau_{s_0} = s, \tau_{a_0} = a)$ of Theorem 3.1. □

### A.2. Proof of Theorem 3.2

*Proof.* Applying ELBO inequality (Song et al., 2021), minimizing Eq. 3 gives us an upper bound on the KL-Divergence of the data distribution $q(\tau^0)$ from the model distribution $p_\theta(\tau^0)$, i.e., $D_{\text{KL}}\left(q(\tau^0) \| p_\theta(\tau^0)\right)$. So, to prove this also gives us an upper bound on $D_{\text{KL}}\left(\pi_{\mathcal{D}_{\text{off}}}(a|s) \| \pi_\theta(a|s)\right)$, we need to show: (1) $D_{\text{KL}}\left(q(\tau^0) \| p_\theta(\tau^0)\right) \geq D_{\text{KL}}\left(\pi_{\mathcal{D}_{\text{off}}}(a|s) \| \pi_\theta(a|s)\right)$; (2) if $p_\theta$ is sufficiently expressive that $\min_\theta D_{\text{KL}}\left(q(\tau^0) \| p_\theta(\tau^0)\right) = 0$, then $\arg\min_\theta D_{\text{KL}}\left(q(\tau^0) \| p_\theta(\tau^0)\right) = \arg\min_\theta D_{\text{KL}}\left(\pi_{\mathcal{D}_{\text{off}}}(a|s) \| \pi_\theta(a|s)\right)$. First, by the KL-Divergence Decomposition and the Chain Rule, we have

$$D_{\text{KL}}\left(q(\tau^0) \| p_\theta(\tau^0)\right) = D_{\text{KL}}\left(q(\{\tau_{s_0}^0, \tau_{a_0}^0, \tau_{s_1}^0, \tau_{a_1}^0, ..., \tau_{s_H}^0, \tau_{a_H}^0\}) \| p_\theta(\{\tau_{s_0}^0, \tau_{a_0}^0, \tau_{s_1}^0, \tau_{a_1}^0, ..., \tau_{s_H}^0, \tau_{a_H}^0\})\right) \tag{14}$$

$$= D_{\text{KL}}\left(q(\{\tau_{s_1}^0, \tau_{a_1}^0, ..., \tau_{s_H}^0, \tau_{a_H}^0\} \mid \{\tau_{s_0}^0, \tau_{a_0}^0\}) \| p_\theta(\{\tau_{s_1}^0, \tau_{a_1}^0, ..., \tau_{s_H}^0, \tau_{a_H}^0\} \mid \{\tau_{s_0}^0, \tau_{a_0}^0\})\right)$$
$$+ D_{\text{KL}}\left(q(\{\tau_{s_0}^0, \tau_{a_0}^0\}) \| p_\theta(\{\tau_{s_0}^0, \tau_{a_0}^0\})\right) \tag{15}$$

$$= D_{\text{KL}}\left(q(\{\tau_{s_1}^0, \tau_{a_1}^0, ..., \tau_{s_H}^0, \tau_{a_H}^0\} \mid \{\tau_{s_0}^0, \tau_{a_0}^0\}) \| p_\theta(\{\tau_{s_1}^0, \tau_{a_1}^0, ..., \tau_{s_H}^0, \tau_{a_H}^0\} \mid \{\tau_{s_0}^0, \tau_{a_0}^0\})\right)$$
$$+ D_{\text{KL}}\left(q(\tau_{s_0}^0) \| p_\theta(\tau_{s_0}^0)\right) + D_{\text{KL}}\left(q(\tau_{a_0}^0 \mid \tau_{s_0}^0) \| p_\theta(\tau_{a_0}^0 \mid \tau_{s_0}^0)\right). \tag{16}$$

Since $\pi_\theta(a|s) = p_\theta(\tau_{a_0}^0 \mid \tau_{s_0}^0)$, and the offline buffer $\mathcal{D}_{\text{offline}} = \{(s, a, r, s')\}$ is sample from the underlying distribution $q(\tau^0)$, yielding the behavior policy $\pi_{\mathcal{D}_{\text{off}}}(a|s) = q(\tau_{a_0}^0 \mid \tau_{s_0}^0)$, by the non-negativity of the KL-Divergence, we obtain

$$D_{\text{KL}}\left(q(\tau^0) \| p_\theta(\tau^0)\right) \geq D_{\text{KL}}\left(q(\tau_{a_0}^0 \mid \tau_{s_0}^0) \| p_\theta(\tau_{a_0}^0 \mid \tau_{s_0}^0)\right) = D_{\text{KL}}\left(\pi_{\mathcal{D}_{\text{off}}}(a|s) \| \pi_\theta(a|s)\right). \tag{17}$$

On the other hand, by the KL-Divergence definition, let $p_\theta^*$ achieves $D_{\text{KL}}\left(q(\tau^0) \| p_\theta^*(\tau^0)\right) = 0$, then we have

$$D_{\text{KL}}\left(q(\tau^0) \| p_\theta^*(\tau^0)\right) = 0 \Leftrightarrow q(\{\tau_{s_0}^0, \tau_{a_0}^0, \tau_{s_1}^0, \tau_{a_1}^0, ..., \tau_{s_H}^0, \tau_{a_H}^0\}) = p_\theta^*(\{\tau_{s_0}^0, \tau_{a_0}^0, \tau_{s_1}^0, \tau_{a_1}^0, ..., \tau_{s_H}^0, \tau_{a_H}^0\}) \quad \text{a.e.} \tag{18}$$

If $q(\{\tau_{s_0}^0, \tau_{a_0}^0, \tau_{s_1}^0, \tau_{a_1}^0, ..., \tau_{s_H}^0, \tau_{a_H}^0\}) = p_\theta^*(\{\tau_{s_0}^0, \tau_{a_0}^0, \tau_{s_1}^0, \tau_{a_1}^0, ..., \tau_{s_H}^0, \tau_{a_H}^0\})$ almost everywhere, then by marginalizing over $\tau_{>0}^0 := \{\tau_{s_1}^0, \tau_{a_1}^0, ..., \tau_{s_H}^0, \tau_{a_H}^0\}$, we get

$$q(\{\tau_{s_0}^0, \tau_{a_0}^0\}) = \int q(\tau_{s_0}^0, \tau_{a_0}^0, \tau_{s_1}^0, \tau_{a_1}^0, ..., \tau_{s_H}^0, \tau_{a_H}^0)d\tau_{>0}^0 = \int p_\theta^*(\tau_{s_0}^0, \tau_{a_0}^0, \tau_{s_1}^0, \tau_{a_1}^0, ..., \tau_{s_H}^0, \tau_{a_H}^0)d\tau_{>0}^0 = p_\theta^*(\{\tau_{s_0}^0, \tau_{a_0}^0\}),$$
(19)

and similarly with $q(\tau_{s_0}^0) = p_\theta^*(\tau_{s_0}^0)$. Therefore

$$p_\theta^*(\tau_{a_0}^0 \mid \tau_{s_0}^0) = \frac{p_\theta^*(\tau_{a_0}^0, \tau_{s_0}^0)}{p_\theta^*(\tau_{s_0}^0)} = \frac{q(\tau_{a_0}^0, \tau_{s_0}^0)}{q(\tau_{s_0}^0)} = q(\tau_{a_0}^0 \mid \tau_{s_0}^0),$$
(20)

i.e., $D_{\mathrm{KL}}\left(q(\tau_{a_0}^0 \mid \tau_{s_0}^0) \parallel p_\theta^*(\tau_{a_0}^0 \mid \tau_{s_0}^0)\right) = 0$. Combining with Eq. 17, we obtain if $\min_\theta D_{\mathrm{KL}}\left(q(\tau^0) \parallel p_\theta(\tau^0)\right) = 0$, then

$$\arg\min_\theta D_{\mathrm{KL}}\left(q(\tau^0) \parallel p_\theta(\tau^0)\right) = \arg\min_\theta D_{\mathrm{KL}}\left(\pi_{\mathcal{D}_{\mathrm{off}}}(a|s) \parallel \pi_\theta(a|s)\right),$$
(21)

i.e., minimizing the objective function in Eq. 3 on the offline buffer $\mathcal{D}_{\mathrm{offline}} = \{(s, a, r, s')\}$ gives us an upper bound on the KL-Divergence of $\pi_{\mathcal{D}_{\mathrm{off}}}(a|s)$ from $\pi_\theta(a|s)$ of Theorem 3.2. $\qquad\square$

### A.3. Proof of Theorem 3.4

*Proof.* From the actor loss function $\mathcal{L}_\pi(\theta) = -\mathbb{E}_{(s,a)\sim\mathcal{D}}[Q_\phi(s, a) \cdot \log \pi_\theta(a|s)]$ in Eq.7, we can express the empirical risk minimization, which typically decomposes into a sum over empirical loss terms $\ell(s, a; \theta)$ and regularizer $r(\theta)$ as follows

$$\theta_{\mathrm{MAP}} = \arg\min_\theta \mathcal{L}_\pi(\theta) \approx \arg\min_\theta \sum_{(s,a)\in\mathcal{D}} [\ell(s, a; \theta)] + r(\theta),$$
(22)

where the loss $\ell(s, a; \theta) = -Q_\phi(s, a) \cdot \log \pi_\theta(a|s)$ and the regularizer $r(\theta) = -\log p(\theta)$. From a Bayesian perspective (Abdolmaleki et al., 2018), these terms correspond to weighted log-likelihoods and a log-prior, respectively; thus, $\theta_{\mathrm{MAP}}$ above is a weighted maximum a-posteriori (MAP) estimate for the weighted log-likelihoods and log-prior as follows

$$\ell(s, a; \theta) = -Q_\phi(s, a) \cdot \log p(a|s; \theta) = -\log p(a|s; \theta)^{Q_\phi(s,a)} \quad \text{and} \quad r(\theta) = -\log p(\theta),$$
(23)

where $p(\theta)$ can be any prior regularizer. Hence, by Bayes' theorem, we can derive the posterior as follows

$$p(\theta|\mathcal{D}) = \frac{p(\mathcal{D}|\theta)p(\theta)}{\int_{\theta'} p(\mathcal{D}|\theta')p(\theta')d\theta'} \approx \frac{\exp(\sum_{(s,a)\in\mathcal{D}}[\log p(a|s; \theta)^{Q_\phi(s,a)}])p(\theta)}{\int_{\theta'} \exp(\sum_{(s,a)\in\mathcal{D}}[\log p(a|s; \theta')^{Q_\phi(s,a)}])p(\theta')d\theta'}$$
(24)

$$= \frac{\exp(\sum_{(s,a)\in\mathcal{D}}[Q_\phi(s, a) \cdot \log p(a|s; \theta)])p(\theta)}{\int_{\theta'} \exp(\sum_{(s,a)\in\mathcal{D}}[Q_\phi(s, a) \cdot \log p(a|s; \theta')])p(\theta')d\theta'}.$$
(25)

Therefore, the exponential of the actor loss $\mathcal{L}_\pi(\theta)$ amounts to an unnormalized posterior. By normalizing it, we obtain

$$p(\theta|\mathcal{D}) = \frac{1}{Z}p(\mathcal{D}|\theta)p(\theta) \approx \frac{1}{Z}\exp(-\mathcal{L}_\pi(\theta)),$$
(26)

with the normalizing constant $Z = \int_{\theta'} p(\mathcal{D}|\theta')p(\theta')d\theta'$. Hence, the Laplace approximation utilizes a second-order Taylor expansion of $\mathcal{L}_\pi(\theta)$ at the mode $\theta_{\mathrm{MAP}}$, resulting in the following approximation

$$\mathcal{L}_\pi(\theta) \approx \mathcal{L}_\pi(\theta_{\mathrm{MAP}}) + \frac{1}{2}\left(\theta - \theta_{\mathrm{MAP}}\right)\left(\nabla_\theta^2 \mathcal{L}_\pi(\theta)|_{\theta=\theta_{\mathrm{MAP}}}\right)\left(\theta - \theta_{\mathrm{MAP}}\right)^\top,$$
(27)

where the disappearance of the first-order Taylor expansion implies that $\theta_{\mathrm{MAP}}$ corresponds to a minimum. Therefore, $Z$ can be expressed as a Gaussian integral

$$Z = \int \exp(-\mathcal{L}_\pi(\theta))d\theta$$
(28)

$$\approx \exp(-\mathcal{L}_\pi(\theta_{\mathrm{MAP}}))\int \exp\left(\theta - \theta_{\mathrm{MAP}}\right)\left(\nabla_\theta^2 \mathcal{L}_\pi(\theta)|_{\theta=\theta_{\mathrm{MAP}}}\right)\left(\theta - \theta_{\mathrm{MAP}}\right)^\top d\theta$$
(29)

$$= \exp\left(-\mathcal{L}_\pi(\theta_{\mathrm{MAP}})\right)(2\pi)^{\frac{|\theta|}{2}}\left(\det(\Sigma)\right)^{\frac{1}{2}}.$$
(30)

As a result, the posterior distribution is approximated with a multivariate Gaussian distribution, i.e., $p(\theta|\mathcal{D}) \approx \mathcal{N}(\theta_{\mathrm{MAP}}, \Sigma)$, where the covariance matrix $\Sigma$ is determined by the Hessian of the posterior, i.e., $\Sigma = \left(\nabla_\theta^2 \mathcal{L}_\pi(\theta)|_{\theta=\theta_{\mathrm{MAP}}}\right)^{-1}$ of Theorem 3.4. $\qquad\square$

# B. Experiments

## B.1. Experimental settings

**Datasets and evaluation metrics**: Following Tarasov et al. (2022), we use the D4RL dataset (Fu et al., 2021; Brockman et al., 2016) for offline-pretraining. D4RL includes multitask datasets where an agent performs different tasks in the same environment, and is collected with mixtures of policies, such as from hand-designed controllers and human demonstrators. In online fine-tuning, the model is fine-tuned by a replay buffer that includes an offline dataset and a cumulated online dataset collected from online interactions. We evaluate the **expected return** (i.e., the expected value of the cumulative reward for each episode) with the online environment. We report the average of expected return in the online phase in all tables, and the expected return for each evaluation in all figures. We select three sets of tasks that are most commonly studied in prior works in offline RL with diffusion (Lu et al., 2025), and also self-design the Frozen-Lake for O2O-RL. Specifically:

- **D4RL MuJoCo Locomotion**: aims to control agents to stand stably or run forward as fast as possible across three sub-environments. For each sub-environment, D4RL includes the **"medium"** (collected from a policy trained to reach approximately 30% expert's performance), **"medium-replay"** (collected during training the medium agent, which includes a mix of early low-quality behaviors & later better behaviors), and **"medium-expert"** (collected from the medium policy & the expert policy) offline dataset. The sub-environments are as follows:

  - **Hopper** has of 11 state features, representing position & velocity of 4 body parts of a hopper (i.e., a one-legged robot). The action has 3 features, representing the torques applied at 3 hinge joints that connect the body parts. The reward has 3 terms to measure how well the robot stands, runs forward, and penalizes for taking too large actions.
  - **Half-Cheetah** consists of 19 state features with position and velocity values of different body parts of a cat-robot. The action has 6 features with the torques applied at six hinge joints that connect the body parts. The reward is based on how well the robot runs forward and is penalized for taking too large actions.
  - **Walker2D** is based on Hopper by adding another set of legs that allow the robot to walk forward instead of hopping. Hence, the reward is the same as the Hopper, while the state has 17 features for the position & velocity of different parts of the body. The action includes 6 features for the torques applied at 6 hinge joints that connect the body parts.

- **ODRL MuJoCo Locomotion**: To examine the real-world transition dynamic, we follow ODRL in Lyu et al. (2024), where halfcheetah-friction, halfcheetah-med-gravity, hopper-kinematic, walker2d-morphology are based on halfcheetah-medium-v2, halfcheetah-medium-replay-v2, hopper-medium-v2, walker2d-medium-v2, respectively, with modified transition dynamics in the online phase with friction, gravity, kinematic, morphology shift levels of 2.0. Details of the shifts are in (Lyu et al., 2022). These attribute modifications allow the simulated robots to significantly change their motion characteristics between the offline and online phases.

- **AntMaze**: aims to navigate a quadrupedal ant-robot through a 2D maze to reach a specific goal location. The 2D maze includes a variety of layouts, e.g., **U-shaped (UMaze)**, **medium-sized**, and **large-sized** mazes. The state consists of 28 features, representing positional and velocity values of different ants' body parts. The action includes 8 features, representing the torques applied at the four-legged robot with eight joints. The reward is the negative Euclidean distance between the achieved position and the desired goal. For each maze layout, the D4RL includes the "play" and "diverse" offline datasets. The **"play" (or non-diverse)** dataset contains trajectories from specific start and goal locations. In contrast, the **"diverse"** dataset is generated by commanding the Ant to travel to random goal locations across the maze.

- **Adroit**: aims to train a robot-hand to perform like a human across four tasks. The action includes from 24 to 30 features depending on the task, representing the angular positions of the Adroit hand joints. The tasks include:

  - **AdroitHandPen-v1** controls the hand to manipulate a pen to achieve a desired goal position and rotation. The state includes 45 features, representing the finger joints' angular position, the palm's pose, and the pose of the real pen and target goal. The reward measures how close the pen is to its target, the similarity of their orientations, and whether the pen is still on the hand.
  - **AdroitHandDoor-v1** controls the hand to open a door with a latch. The state has 39 features for the finger joints' angular position, palm's pose, and latch and door's information. The reward measures how close the palm is to the door handle, the current door hinge angular position v.s. open door state, velocity penalty, and how the door hinge is opened more than some radians.
  - **AdroitHandHammer-v1** controls the hand to hammer a nail into a board. The state has 46 features for finger joints' angular position, palm's pose, hammer & nail's pose, and external forces on the nail. The reward measures how close the palm is to the hammer, hammer's head from nail's head, nail's head from/to board, velocity penalty, and how far the hammer is lifted.

- **AdroitHandRelocate-v1** controls the hand to pick up a ball & move it to a target location. The state has 39 features, representing the ball's & target's finger joints' angular position, the palm's pose, and kinematic information. The reward measures how close the palm is to the ball, whether the ball is lifted from the table, and the ball's distance to its target.

- **Frozen-Lake**: aims to navigate an agent through a 2D frozen-lake map to reach a specific goal location. The state is the agent's position on a 2D map. The action is whether to move up, down, left, or right. The reward is 1 if the agent reaches the goal location and 0 if not, or is terminated by reaching the hole or limiting the episode. There are slippery sections on the map that may cause the robot to move in an undesirable direction. We collect an offline dataset from samples acquired by the $\epsilon$-greedy algorithm. Notably, to highlight the state-transition shift, we follow Bui et al. (2026) and change the slippery level (i.e., the probability that the agent will slip, moving in a perpendicular direction to the intended direction) from $1/4$ during offline data collection to $1/3$ during online fine-tuning. As a result of this setting, we can ensure that the state-transition distribution $\mathbb{P}(s'|s,a)$ also changes accordingly.

**Baseline and hyper-parameters details:** In our experiments, we select baseline and hyper-parameters to follow exactly the high-quality O2O-RL benchmark from Tarasov et al. (2022). Note that the O2O-RL benchmark defines the reward as the default instead of the spare reward in Zhou et al. (2025). For a fair comparison, all diffusion models are trained with the same offline dataset in pre-training and the same replay buffer in online fine-tuning. Based on two modified critics (i.e., IQL (Hansen-Estruch et al., 2023) and Cal-QL (Nakamoto et al., 2023)), we compare with the following diffusion-based actor models and their UQ variants:

- **IQL** (Tarasov et al., 2022): the loss $\mathcal{L}_{TD}(\phi)$ in Eq. 1 is extended to avoid querying out-of-sample (unseen) actions by considering fitted $Q$ evaluation with a SARSA-style objective, which aims to learn the value of the behavior policy, defined by the following critic loss

$$\mathcal{L}_Q(\phi) = \mathbb{E}_{(s,a,s',a')\sim\mathcal{D}}\left[\{(r(s,a) + Q_\phi(s',a')) - Q_\phi(s,a)\}^2\right].$$

- **Cal-QL** (Nakamoto et al., 2023): the loss $\mathcal{L}_{TD}(\phi)$ in Eq. 1 is extended to learns conservative value functions that are constrained to be larger than the value function of a reference policy to facilitate fast online fine-tuning, defined by the following critic loss

$$\mathcal{L}_Q(\phi) = \mathbb{E}_{s\sim\mathcal{D},a\sim\pi}\left[\max\left(Q_\phi(s,a), \mathbb{E}_{a\sim\mu}\left[Q^\mu(s,a)\right]\right)\right] - \mathbb{E}_{(s,a)\sim\mathcal{D}}\left[Q_\phi(s,a)\right],$$

where $Q^\mu(s,a)$ is the calibrated $Q$-function of the learned $Q$-function $Q_\phi^\pi$ w.r.t. a reference policy $\mu$, i.e., $\mathbb{E}_{a\sim\mu}\left[Q_\phi^\pi(s,a)\right] \geq \mathbb{E}_{a\sim\mu}\left[Q^\mu(s,a)\right], \forall s \in \mathcal{D}$.

- **SO2** (Zhang et al., 2024): the loss $\mathcal{L}_{TD}(\phi)$ in Eq. 1 is extended with ensemble Q-value to expand the action distribution used to estimate the target Q-value, yielding a smoother estimate, defined as follows

$$Q_{\phi_i}(s,a) \leftarrow r + \gamma\left(\hat{Q}_{\phi_i}(s',a'+\epsilon) - \beta\log\pi(a'|s')\right),$$

where $a' \sim \pi(\cdot|s')$, $\epsilon \sim clip(\mathcal{N}(0,\sigma), -c, c)$, $i = 1, \cdots, N_{\text{ensemble}}$, $\epsilon$ is the Gaussian noise bounded by c, and $\phi_i$ is the parameter of $i$-th $Q$ model in the $N_{\text{ensemble}}$. Note that we use the **SO2+OFF2ON** setting, where SO2 is combined with OFF2ON (Lee et al., 2022) to encourage the use of near-on-policy samples from the offline dataset.

- **EDIS** (Liu et al., 2024): use diffusion planner in Sec. 2.2, with energy guider $\varepsilon(\tau) = \frac{q(\tau^0)^{\text{online}}}{q(\tau^0)^{\text{offline}}}$ (trained from $\mathcal{D}_{\text{offline}} \cup \mathcal{D}_{\text{offline}}$) to augment trajectory data that conforms to the online distribution by $p_\theta(\tau)^{\text{online}} \propto p_\theta(\tau)^{\text{offline}} e^{-\varepsilon(\tau)}$. The augmented data from $p_\theta(\tau)^{\text{online}}$ is added to the experience replay buffer to fine-tune another actor-critic model. The trajectory length is $H = 8$, and the denoising timestep is $N = 32$.

- **Diffuser**: We extend from Janner et al. (2022) in offline RL to our O2O-RL, where we use diffusion planner in Sec. 2.2 to output a trajectory. The Diffuser is pre-trained on offline dataset $\mathcal{D}_{\text{offline}}$ and fine-tuned with the replay buffer $\mathcal{D}_{\text{offline}} \cup \mathcal{D}_{\text{offline}}$. The action is obtained by using our policy extraction in Eq. 4. Following (Lu et al., 2025), the trajectory length is $H = 8$ and denoising timestep is $N = 20$.

- **Diff-QL**: We extend from Wang et al. (2023b) in offline RL to our O2O-RL, where we use the Diff-QL policy to output an action conditioned on given input state. The Diff-QL policy is pre-trained on offline dataset $\mathcal{D}_{\text{offline}}$ and fine-tuned with the replay buffer $\mathcal{D}_{\text{offline}} \cup \mathcal{D}_{\text{offline}}$. The number of denoising timesteps is $N = 8$.

- **Diff-QL w LA$\mathcal{L}_\pi$**: is an extension of Diff-QL above, where we additionally combine with our Laplace approximation in Eq. 6 to obtain epistemic UQ. The number of MC samples in policy sampling in Eq. 7 is $M = 5$.

- **DACER**: We extend from DACER in online RL (Wang et al., 2024) to our O2O-RL, where we leverage the Diff-QL above to get its UQ in the online phase by using a Gaussian mixture model (GMM) to fit the policy distribution and compute the entropy of this action distribution. By using GMM with 3 number of Gaussian distributions, for each online time step, DACER requires the diffusion policy samples 200 times to get different actions to obtain the final UQ with entropy, causing a very high latency in Fig. 2.

- **Ours method (DUAL)**: Since we extract our policy from Diffuser Planner in EDIS (Liu et al., 2024), we run progressive distillation (Salimans and Ho, 2022) to reduce the number of denoising timestep from $N = 32$ to $N = 1$, and the trajectory length is $H = \{1, 8\}$. The number of MC samples in policy sampling in Eq. 7 is $M = 5$.

**Source code and computing system.** The source code to reproduce our results is available at this GitHub link. We train our model on two single GPUs: NVIDIA A100-PCIE-40GB with 8 CPUs: Intel(R) Xeon(R) Gold 6248R CPU @ 3.00GHz with 8GB RAM per each, and require 32GB available disk space for storage. We test our model on three different settings, including (1) a single GPU: NVIDIA Tesla K80 accelerator-12GB GDDR5 VRAM with 8-CPUs: Intel(R) Xeon(R) Gold 6248R CPU @ 3.00GHz with 8GB RAM per each; (2) a single GPU: NVIDIA RTX A5000-24564MiB with 8-CPUs: AMD Ryzen Threadripper 3960X 24-Core with 8GB RAM per each; and (3) a single GPU: NVIDIA A100-PCIE-40GB with 8 CPUs: Intel(R) Xeon(R) Gold 6248R CPU @ 3.00GHz with 8GB RAM per each.

## B.2. Additional results

**Ablation study with $\lambda$:** We test the robustness of the shift-awareness hyper-parameter $\lambda$ across different values, including $\{0.0, 0.1, 0.25, 0.5, 0.75, 1.0\}$. Tab. 5 shows its average online return on D4RL hopper-medium (i.e., no transition shift), and O4RL hopper-friction environment (i.e., transition shift). Regarding when there is no transition shift, we observe that our results are stable across $\lambda = (0.1, 0.5)$ in most D4RL tasks, such as the hopper-medium in Tab. 5. This is because D4RL tasks focus on the policy shift rather than the transition shift. However, when the transition shift occurs, we observe that without shift-awareness (i.e., $\lambda = 0$), the online return drops significantly (e.g., hopper-friction in Tab. 5, Frozen-Lake in Fig. 3), showing that the shift-awareness component is crucial when there are transition shifts.

*Table 5.* Average online return across different $\lambda$ in Eq.8 on the D4RL hopper-medium and ODRL hopper-friction environments.

| $\lambda$ | 0.0 | 0.1 | 0.25 | 0.5 (default) | 0.75 | 1.0 |
|---|---|---|---|---|---|---|
| hopper-medium | $87.30 \pm 3.44$ | $88.07 \pm 3.36$ | $88.50 \pm 3.36$ | $88.23 \pm 3.42$ | $87.30 \pm 3.64$ | $86.81 \pm 4.00$ |
| hopper-friction | $35.12 \pm 3.80$ | $42.11 \pm 3.40$ | $47.96 \pm 3.38$ | $48.00 \pm 3.44$ | $49.58 \pm 3.33$ | $47.25 \pm 3.60$ |

**Ablation study on how the quality of the logging policy affects the performance**: We evaluate our model performance across different offline logging policies with hopper-medium-v2 and hopper-medium-expert-v2, with 33%, 67%, and 100% dataset size in Tab. 6. In the first row, we measure how many online steps our model needs to reach the same performance at 90 average online return. We observe that hopper-medium-expert-v2 has very good logging policies, and the model can achieve a 90 return without needing online iterations. Meanwhile, the logging policies in hopper-medium-v2 are worse, causing around 1.1M online steps to be required to reach the same performance. In the last two rows, we compare our model performance between with and without the policy extraction from the diffusion planner. We can see that if the model is only trained on a very sub-optimal offline dataset with low-quality policies (e.g., 33% hopper-medimum), the difference in online return between with and without planner-prior extraction is quite small. This implies that the distillation from the diffusion planner may not be really effective for the policy if trained with very low-quality policies.

*Table 6.* Average online return across different offline logging policies budget with hopper-medium-v2 and hopper-medium-expert-v2.

| Method | 33% ho-medium | 33% ho-expert | 67% ho-medium | 67% ho-expert | ho-medium | ho-expert |
|---|---|---|---|---|---|---|
| DUAL (#steps) $\rightarrow$ 90 | 1.4M | 0.8M | 1.3M | 0.4M | 1.1M | 0 |
| DUAL (Ours) | $71.10 \pm 3.66$ | $74.28 \pm 4.69$ | $78.10 \pm 3.51$ | $86.34 \pm 4.63$ | $88.23 \pm 3.42$ | $115.33 \pm 4.58$ |
| w/o planner-prior extraction | $70.09 \pm 3.92$ | $71.76 \pm 4.90$ | $72.98 \pm 3.92$ | $80.96 \pm 4.88$ | $82.08 \pm 3.81$ | $111.12 \pm 4.60$ |

**Ablation study on how the size of the offline-dataset affects the online budget**. From Tab. 6, we also observe that the size of the offline dataset significantly affects the online budget. This is because diffusion models often require a certain amount of data to learn high-dimensional and complex data distributions effectively. Notably, we observe that if the offline-dataset

includes high-quality behavior policies (e.g., hopper-medium-expert-v2), the offline-dataset's size affects more significant to the online performance. That said, if the offline-dataset includes lower-quality behavior policies, the effectiveness of the offline-dataset's size will have less significance to the online performance (e.g., hopper-medium-v2).

**Training cost**: Compared with EDIS baseline (another diffusion framework for O2O-RL), in antmaze-umaze-diverse-v2, EDIS needs around 3 hours to run with NVIDIA A-100. Meanwhile, our method needs about 4 hours, including 3 hours similar to EDIS (with diffusion planner pre-training (L3-4 in Alg. 1), actor-critic training (L6-7)), 55 mins for the distillation step, and 5 mins for Laplace approximation. Notably, our Laplace approximation step (L-9) causes very minor training time when compared to the distillation step (L-5), as we only do the last-layer Laplace approximation in Eq. 6.

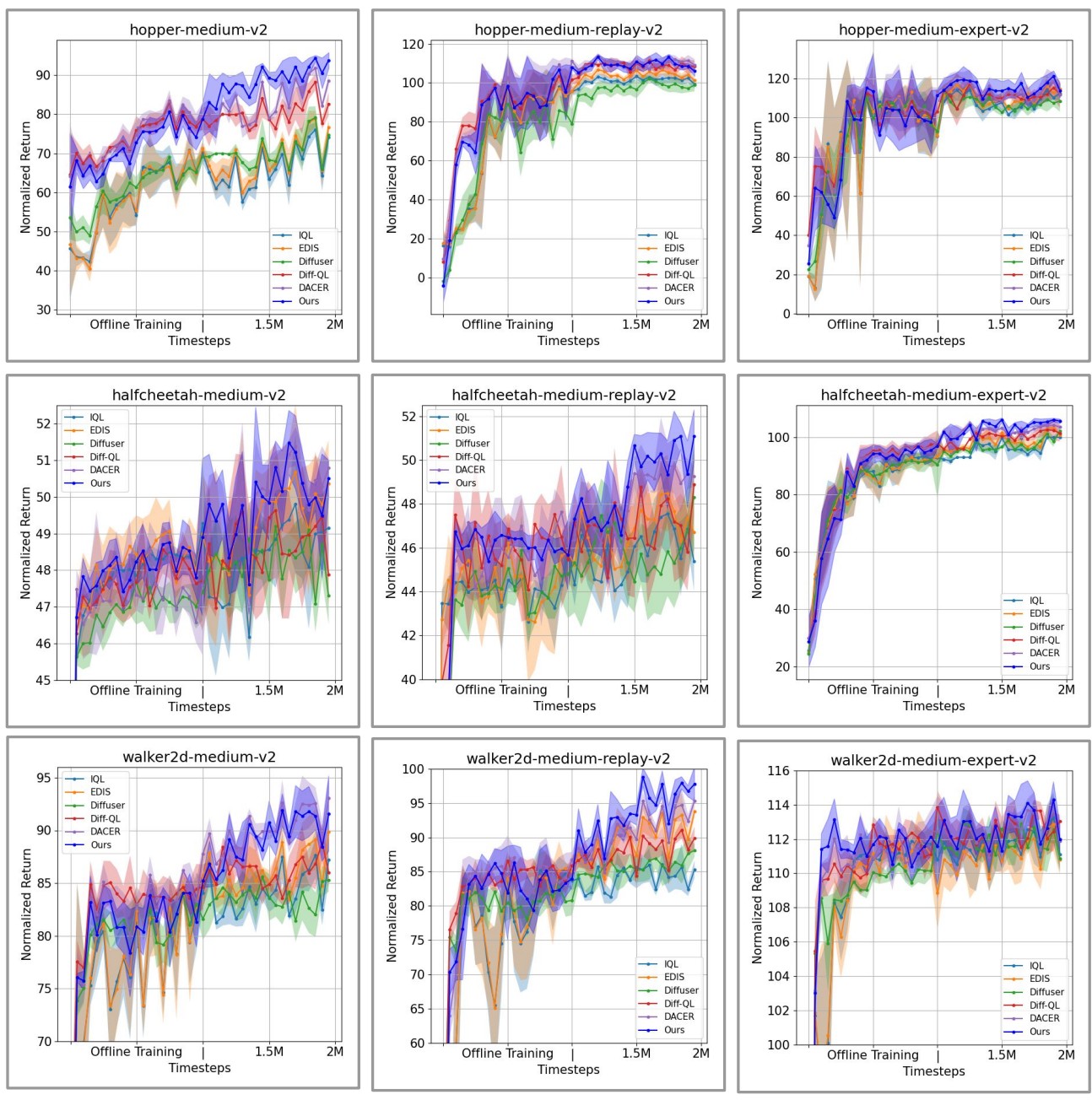

*Figure 4.* Offline-to-Online Benchmark with diffusion baselines on MuJoCo Locomotion environments. This is a simple short-term planning task, where the goal is to control agents (e.g., humanoids, quadrupeds) to stand stably or run faster. Overall, all methods cover quickly after a few training epochs. That said, diffusion policy methods like Diff-QL and DACER are still better than the non-diffusion IQL baseline. Notably, our method also leverages the diffusion policy with a high-quality epistemic UQ, which can balance the exploration and exploitation, resulting in an improvement in expected return in the online phase in some tasks, such as in "medium-v2" and "walker2d-medium-replay-v2".

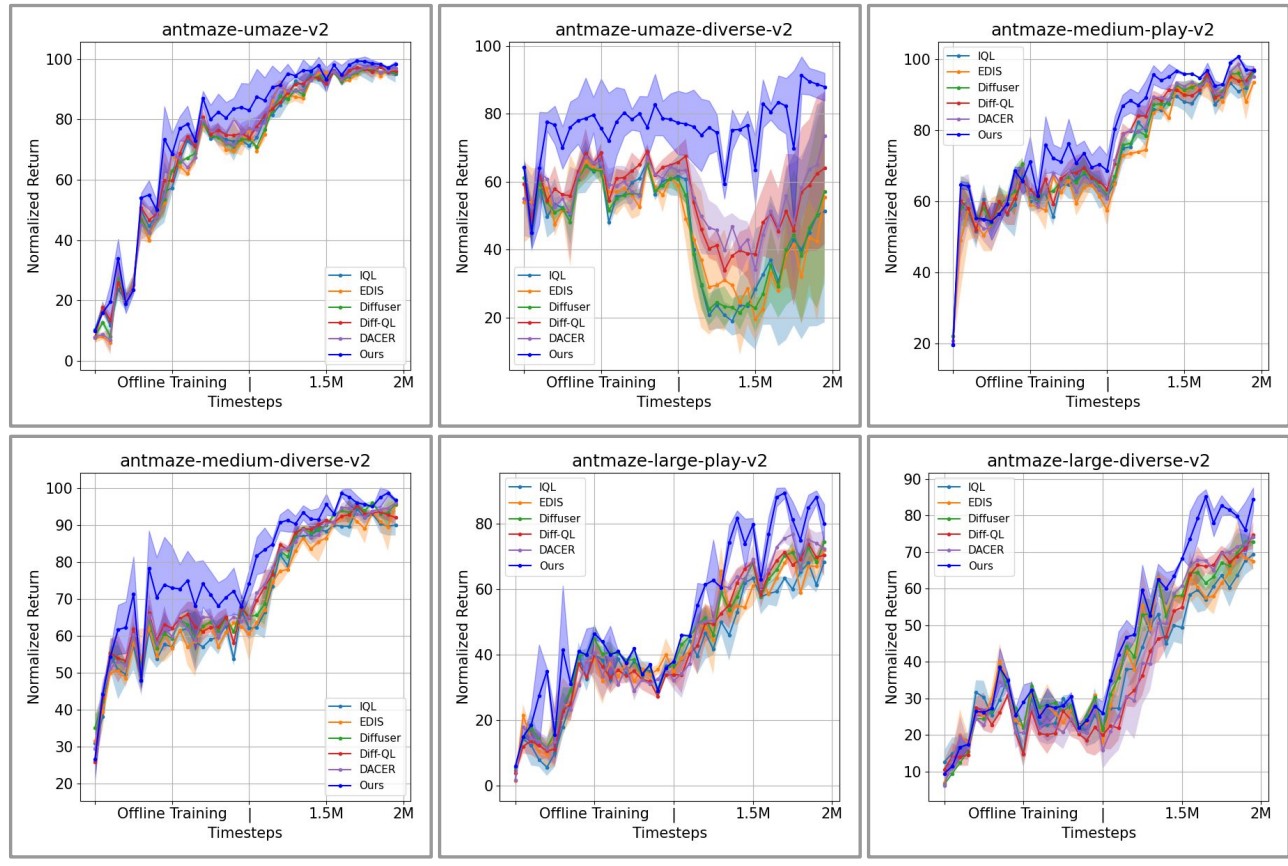

*Figure 5.* O2O-RL Benchmark with diffusion baselines on AntMaze. Our method outperforms significantly other baselines in this complex long-term planning task, where the goal is to train a four-legged ant agent to navigate across different 2D mazes to reach a target location.

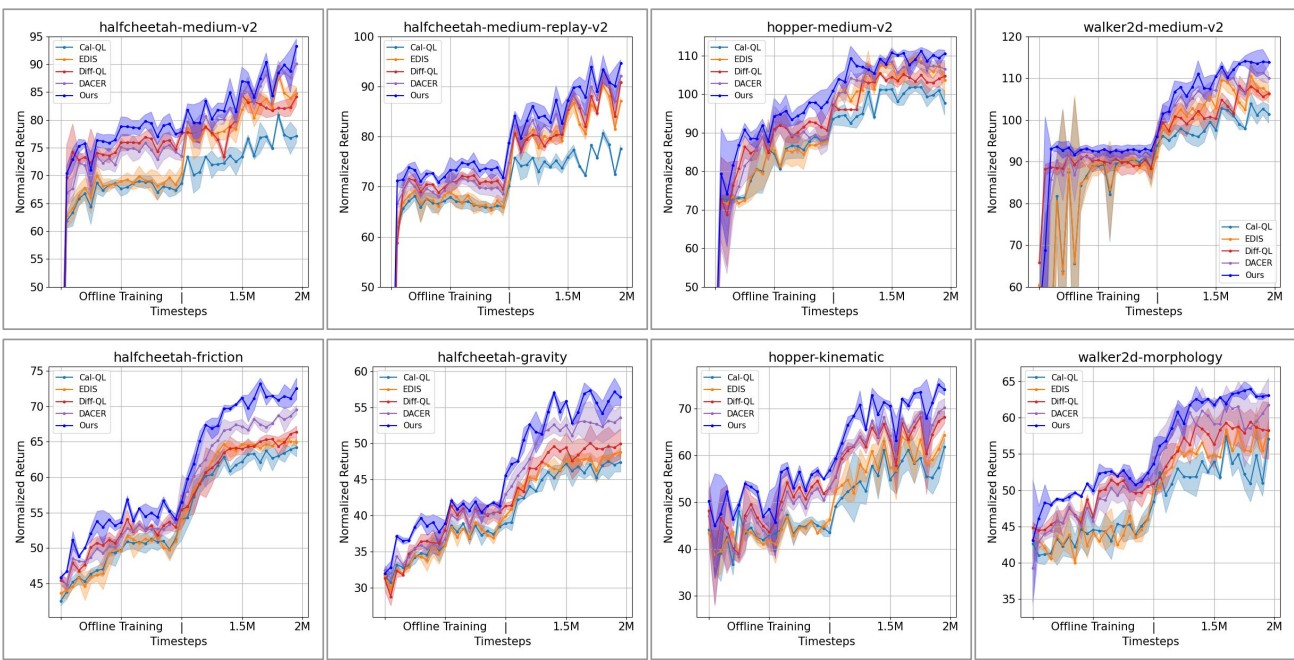

*Figure 6.* Offline-to-Online Benchmark with Cal-QL based methods on D4RL and ODRL MuJoCo environments.

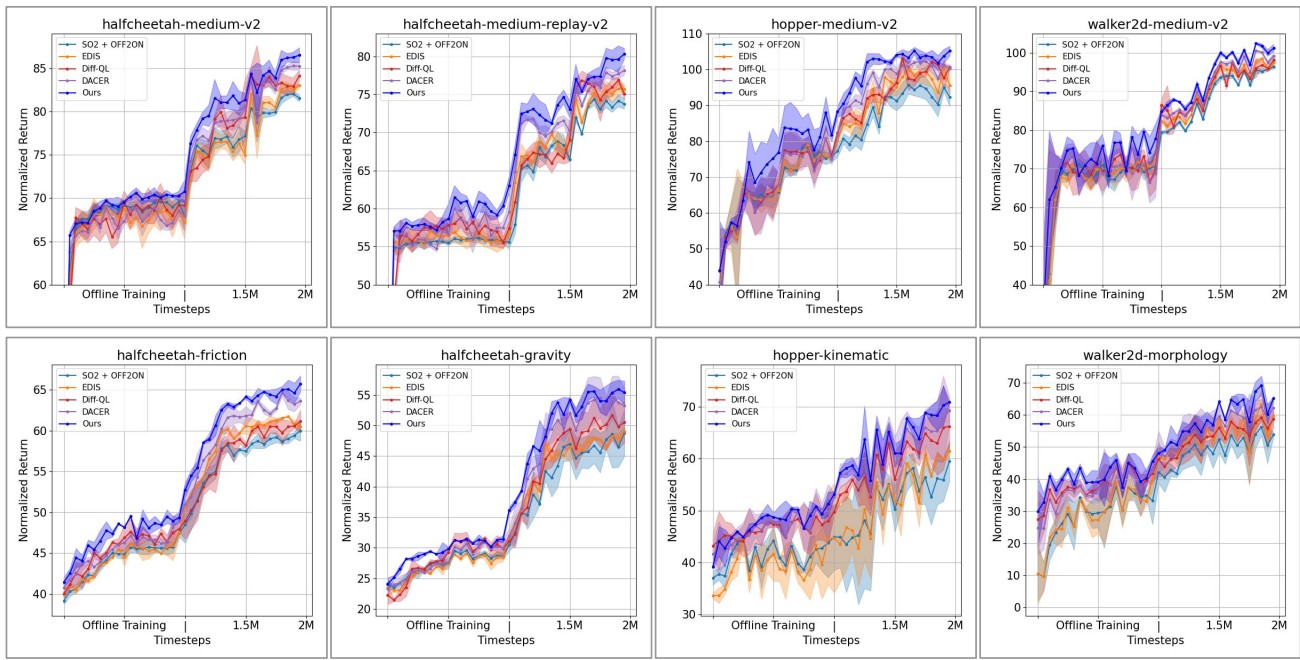

*Figure 7.* Offline-to-Online Benchmark with SO2+OFF2ON-based methods on D4RL and ODRL MuJoCo environments.

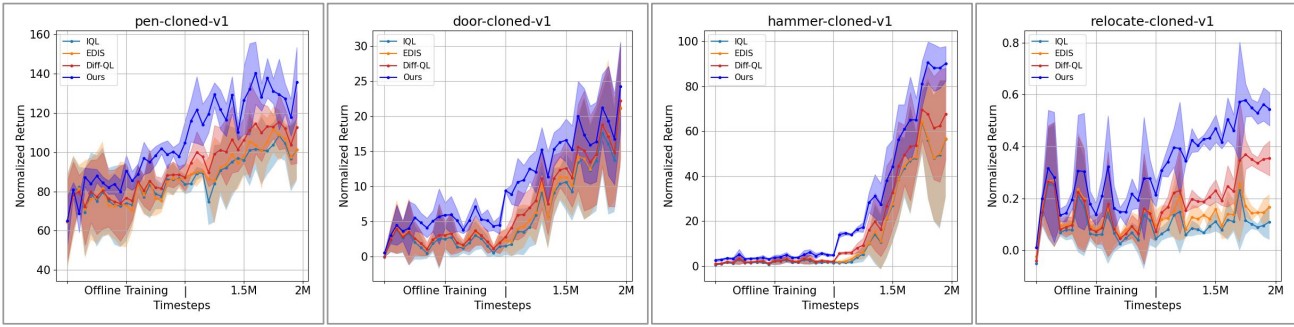

*Figure 8.* Offline-to-Online Benchmark with IQL-based methods on Adroit environments.

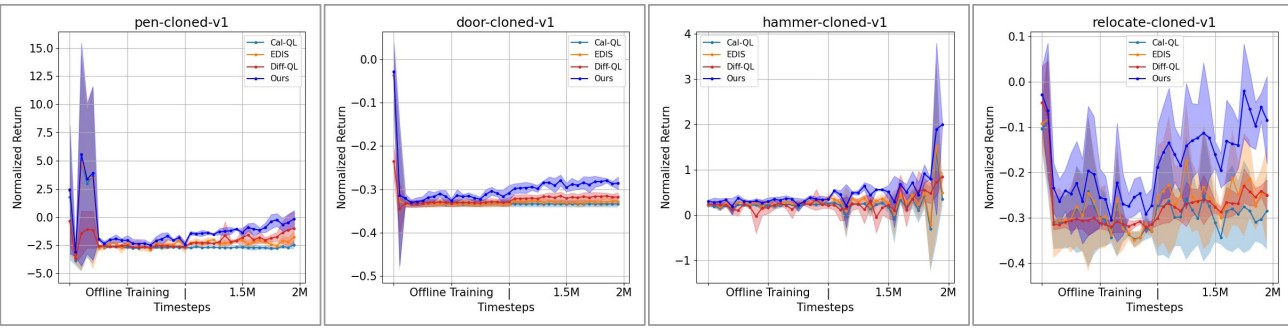

*Figure 9.* Offline-to-Online Benchmark with Cal-QL-based methods on Adroit environments.

