# OpenReview forum: "Efficient and Uncertainty-Aware Diffusion Framework for Offline-to-Online Reinforcement Learning"
_ICML.cc/2026/Conference — ICML 2026 regular_

### Official Review · Reviewer_Gss5 · 2026-03-01

**Soundness:** 2
**Presentation:** 3
**Significance:** 2
**Originality:** 2
**Overall Recommendation:** 2
**Confidence:** 4

**Summary:**

This paper introduces DUAL, a Diffusion Uncertainty-Aware Actor-Critic framework designed to enhance Offline-to-Online Reinforcement Learning. DUAL addresses the challenge of distribution shift by  employing uncertainty quantification techniques, i.e. Laplace approximation and state-shift detection, to balance exploration and exploitation during online fine-tuning.

**Compliance With Llm Reviewing Policy:**

Affirmed.

**Final Justification:**

## Post-Rebuttal

After careful and thorough consideration of the paper and the authors’ rebuttal, I believe there remain issues:

1. I am confident that the proof of Theorem 3.4 is incorrect, which raises serious concerns about the validity of the method. Although the authors attempted to address this during the discussion, the concern remains unresolved. I also responsed this during the discussion. Actually, even at a glance, it is unclear how one can derive a posterior probability conditioned solely on the dataset $\mathcal{D}$, given that new actions are introduced during training by the learned policy $\pi$.

2. The response in rebuttal 5.2 regarding the claimed advantages over directly training the behavior policy and transition model is not convincing. Since the diffusion planner is essentially trained via supervised learning, the resulting policy closely resembles behavior cloning, which is also supported by Theorem 3.2. The authors argue that their approach can avoid local optima; however, this claim lacks theoretical justification. Furthermore, no empirical evidence is provided to substantiate this advantage. In contrast, directly training the components would likely reduce computational cost and could be made more efficient through techniques such as progressive distillation.

3. The reported performance does not even surpass that of purely offline RL methods in some tasks, as previously noted.

Given these unresolved theoretical and empirical concerns, I lean toward a reject recommendation.

**Key Questions For Authors:**

Questions:

1. The planner is an unconditional diffusion model, while the behavior policy and transition dynamics are conditioned on $s$ and $(s,a)$, respectively. How is distillation performed from the planner without additional training? If distillation does require training, what is the advantage over directly training the behavior policy and transition model on the dataset, which would avoid the computational overhead associated with sampling from the planner?

2. For which specific tasks were the optional steps in the algorithm employed?

3. See weaknesses.

PS:

A typo in the algorithm, line 12: $a_{t}$ should be $a_{t-1}$.

**Limitations:**

More comparison with other O2O methods should be incorporated.

**Strengths And Weaknesses:**

Strengths:

1. The paper is well-structured, making it accessible to readers.
2. The experimental evaluation is comprehensive.

Weaknesses:

1. The primary contribution, i.e. epistemic uncertainty quantification for online exploration, appears to be an incremental extension of the method proposed in Generative Uncertainty in Diffusion Models [1].

2. As shown in Figure 4, performance gains after 1 million online interaction steps are marginal across most tasks, with the exception of hopper-medium and walker2d-medium-replay. This suggests that the proposed exploration strategy may not be consistently effective.

3. The experimental comparison omits several relevant offline-to-online RL methods, including CalQL [2], SO2 [3], CFDG [4], and WSRL [5]. Notably, the proposed method appears to underperform these baselines on MuJoCo (compared to CalQL, SO2, and CFDG) and on Antmaze (compared to WSRL), which weakens the empirical claims of superiority.

4. The experiment Lacks the sensitivity analysis of $\lambda$. Additionally, the ablation study is limited to only one task per domain (MuJoCo and Antmaze); including more diverse tasks such as halfcheetah-medium and hopper-medium-replay would strengthen the analysis. Furthermore, the ablation study on the hopper-medium task appears to be missing several ablations related to the transition error temr in the exploration.


[1] Jazbec, M. et al. Generative Uncertainty in Diffusion Models, UAI.

[2] Nakamoto, M. et al. Cal-QL: Calibrated Offline RL Pre-Training for Efficient Online Fine-Tuning, NeurIPS.

[3] Zhang, Y. et al. A Perspective of Q-value Estimation on Offline-to-Online Reinforcement Learning,  AAAI.

[4] Huang, X. et al. Offline-to-Online Reinforcement Learning with Classifier-Free Diffusion Generation, ICML

[5] Zhou, Z. et al. Efficient Online Reinforcement Learning Fine-Tuning Need Not Retain Offline Data, ICML

---

> ### Author Rebuttal · Authors · 2026-03-31
>
> We thank the reviewer for valuable comments. We address your concerns below.
>
> **1. Epistemic UQ contribution appears to be incremental from [1].** Although we shared the Laplace framework with [1], the way we utilize Laplace approximation is distinctly separate from [1], and our method contributes to a better epistemic UQ quality for diffusion RL. Specifically, [1] applies Laplace with ELBO loss, which is not theoretically valid for epistemic UQ (Rmk.3.3). To address this issue, we apply Laplace with our proposed actor loss in Eq.7-8. Thm.3.4 confirms our approach is a valid estimate of epistemic UQ, helping to better balance exploration and exploitation in RL experiments.
>
> **2. Performance gains after 1M online steps are marginal in Fig.4. Consistently effective?** In some tasks in Fig.4, we observe that the logging data (offline + online) is quite good and saturated, leading to all methods converging quickly after 1M online steps. That said, our method converges faster than others (i.e., before 1M online steps). This is crucial in O2O-RL because the goal is to minimize costly online interactions. Regarding consistent effectiveness, our method consistently achieves a higher return than others, with significant tests in Tab.1.
>
> **3. Comparisons with [2,3,4,5]. Underperformance on MuJoCo, Antmaze?** Our work focuses on designing an efficient diffusion actor, and it is compatible with and can be implemented on top of all [2,3,4,5]. We also compared with CalQL [2] on Adroit. Regarding CFDG (EDIS's extension), [4] hasn't published its source code to extend its results, but we compared with EDIS.
>
> In MuJoCo, our lower result is due to the use of IQL for the critic model and the absence of data augmentation. Our method can certainly use advanced critic models (e.g., CalQL, SO2) or data augmentation (CFDG, EDIS) to improve performance. We report with CalQL [2], SO2 [3] on halfcheetah-medium-v2:
>
> |Method|CalQL|Ours+CalQL|SO2|Ours+SO2|
> |---|---|---|---|---|
> |Avg online return|93.11±1.91|97.26±1.20|95.93±3.42|99.02±1.97|
>
> In Antmaze, our results are not comparable to WSRL because [5] uses another critic with a sparse reward of $5$ at goal \& $-5$ otw. We follow CORL benchmark, which uses IQL with the default reward in gym ($1$ at goal \& $-1$ otw). Thanks for your suggestion. Since [5] shows a framework to discard offline data without destabilizing online fine-tuning, we can certainly combine with WSRL in our revision.
>
> **4. Ablation study: more diverse tasks would strengthen the analysis. Lacks analysis of $\lambda$, hopper-medium transition error term.** Thanks for your valuable suggestion. We added an ablation study using hopper-medium-replay in off-dynamics environments (i.e., transition shift). We refer this to our response to R-Qcy5. Regarding $\lambda$, we observe our results are robust between $\lambda=(0.1,0.5)$ in most tasks with D4RL, except the Frozen-Lake in Fig.3. This is because D4RL tasks focus on policy shift, not transition shift. As suggested, we report the results with the hopper-medium-v2 in D4RL for different $\lambda$:
>
> |$\lambda$|0.0|0.1|0.25|0.5 (default)|0.75|1.0|
> |---|---|---|---|---|---|---|
> |Avg online return|87.30±3.44|88.07±3.36|88.50±3.36|88.23±3.42|87.33±3.64|86.81±4.00|
>
> **5.1. How is distillation performed from the planner without additional training?** We are unsure what the reviewer means by "additional training"; any clarification would be appreciated. We pre-train the diffusion planner with a trajectory input (minimizing the KL w.r.t. joint distribution $(s,a,s')$ of the model from data distribution), then we do progressive distillation to get a faster model. Since the divergence between joint distribution is minimized, both the conditional on $s$ and on $(s,a)$ are minimized (e.g., Thm.3.2). In inference time, we therefore set the first state r.v. in the trajectory input to a specific state value, then get the corresponding action and next state output from the diffusion planner.
>
> **5.2. What is the advantage over directly training the behavior policy and transition model?** Our method has two main benefits. First, directly training the behavior policy on the dataset causes the diffusion policy to get stuck at local optimals by only looking at the next optimal action. Our proposed method, i.e., training a diffusion planner and then extracting a policy from it, can help avoid these local optima, as the diffusion planner is trained with a longer trajectory horizon. Second, we need only one diffusion model, which may save computational cost over two separate diffusions for policy and transition models. We agree that a longer trajectory edges computational cost, yet this is minor compared to the denoising steps of two separated models, which is a main bottleneck in diffusion.
>
> **6. Which tasks were the optional steps employed?** We only use the optional step in the ablation study in Fig.3.(b).
>
> **7. Typo in L-12 algorithm: $a_t$ should be $a_{t-1}$.** Thanks, we fixed this typo.

---

> > ### Author Rebuttal · Reviewer_Gss5 · 2026-04-02
> >
> > The first response raises **significant concerns** regarding the overall correctness of the proposed epistemic UQ. It suggests that the authors may have misunderstandings about the role of ELBO in [1]. Specifically, Remark 3.3 appears to introduce two misleading interpretations for readers:
> > (1) The ELBO in Eq. (5) is actually attributed to the planner network rather than the policy. The claim about ELBO obviously distort the original meaning in [1];
> > (2) Theorem 3.4 is claimed to follow by directly replacing the ELBO objective with the actor loss in Eq. (7), which is not obviously justified.
> >
> > These issues motivated me a closer examination of the proof of Theorem 3.4 (I overlooked this point in my original comments.). A key concern arises in the derivation of Eq. (24). The actor loss $\mathcal{L} _{\pi}(\theta) = - \mathbb{E} _{s,a} [Q^{\pi _{\rm old}} (s,a) \cdot \log \pi _{\theta} (a \mid s)]$ **is explicitly weighted by Q-value**, and therefore does not correspond to a standard negative log-likelihood. As a result, the identification $\mathcal{L} _{\pi}(\theta) = -  \log \pi _{\theta} (a \mid s)$ in Eq. (24) is not valid. Without a well-defined likelihood model, **the claim that MAP estimate is not substantiated**. Even at a glance, it is unclear how $- \mathbb{E} _{s,a} [Q^{\pi _{\rm old}} (s,a) \cdot \log \pi _{\theta} (a \mid s)]$ can be reduced to a form such as $-  \log \pi _{\theta} (a \mid s)$.
> > Relatedly, the assertion $0 = - \log p(\theta)$, which implicitly assumes a constant prior, is also questionable.  Additionally, there is a sign inconsistency in the covariance derivation: since the Hessian should be positive definite, the covariance matrix in the Laplace approximation should not include a negative sign. Overall, Eq. (24) does not appear to follow from the preceding arguments, which raises concerns that the proof may be fundamentally flawed rather than a minor technical oversight.
> >
> > On the empirical side, the ablation and sensitivity results on hopper-medium raise further questions:
> > (1) setting the $\lambda=0$ appears to have minimal impact on performance, suggesting limited contribution from the proposed component;
> > (2) the reported online performance of optimal $\lambda$ does not even surpass some strong offline RL baselines, such as DTQL.
> >
> > I sincerely look forward to the authors’ clarification on these points and will update my final assessment accordingly. I also encourage the Area Chair to carefully evaluate the theoretical claims, as they appear questionable.

---

> > > ### Author Response · Authors · 2026-04-03
> > >
> > > **1. The role of ELBO in Jazbec's work [1].** Thanks for your constructive comment. Since the policy $\pi_\theta$ is extracted from the diffusion planner $p_\theta(\tau)$, Rmk.3.3 just wants to explain why we don't apply ELBO-based Laplace [1] to approximate the posterior distribution $p(\theta|\mathcal{D})$, because ELBO loss is not interpreted as a log-likelihood or weighted log-likelihood distribution. We will clarify more about this in our revision.
> > >
> > > **2. Clarification for proof in Thm.3.4.** We agree that $L_\pi(\theta) = -\log p(a|s;\theta)$ in Eq.24 is not valid, since we already used $L_\pi(\theta)$ to denote the entire loss in Eq.7. We would like to fix $L_\pi(\theta) = -\log p(a|s;\theta)$ in our Eq.24 to $\ell(s,a;\theta) = -Q^{\pi _ {\mathrm{old}}}(s,a)\cdot \log p(a|s;\theta)$, and show how we can reach to Eq.25 in proof of Thm.3.4 below:
> > >
> > > ---
> > > From the actor loss $\mathcal{L}\_{\pi}(\theta) = -\mathbb{E}\_{s \sim \mathcal{D},a \sim \pi }[Q^{\pi _ {\mathrm{old}}}(s,a) \cdot \log\pi_\theta(a|s)]$ in Eq.7, we can express the empirical risk minimization, which typically decomposes into a sum over empirical loss terms $\ell(s,a;\theta)$ and regularizer $r(\theta)$ as follows
> > >     $$\theta_{\text{MAP}}= \arg\min_\theta \mathcal{L}\_{\pi}(\theta) \approx \arg\min_\theta \sum_{(s,a) \in S \sim \mathcal{D}\times \pi}[\ell(s,a;\theta)] + r(\theta),$$
> > >     where $\ell(s,a;\theta)=-Q^{\pi _ {\mathrm{old}}}(s,a)\cdot \log \pi_\theta(a|s)$ and $r(\theta)=-\log(1)$. From a Bayesian perspective, these terms correspond to weighted log-likelihoods and a log-prior, respectively; thus, $\theta_{\text{MAP}}$ above is a maximum a-posteriori estimate for the following weighted log-likelihoods and log-prior
> > >     $$\ell(s,a;\theta) = -Q^{\pi _ {\mathrm{old}}}(s,a)\cdot \log p(a|s;\theta) = - \log p(a|s;\theta)^{Q^{\pi _ {\mathrm{old}}}(s,a)} \quad \text{and} \quad r(\theta) = -\log p(\theta),$$
> > >     where $p(\theta)$ is assumed to be uniformly distributed with density $1$. Hence, by Bayes' theorem, we can derive the posterior
> > >     \begin{align}
> > >         p(\theta|\mathcal{D}) = \frac{p(\mathcal{D}|\theta) p(\theta)}{\int_{\theta'} p(\mathcal{D}|\theta') p(\theta')d\theta'}
> > >         &\approx \frac{\exp(\sum_{(s,a) \in S \sim \mathcal{D}\times \pi}[\log p(a|s;\theta)^{Q^{\pi _ {\mathrm{old}}}(s,a)}]) p(\theta)}{\int_{\theta'} \exp(\sum_{(s,a) \in S \sim \mathcal{D}\times \pi}[\log p(a|s;\theta')^{Q^{\pi _ {\mathrm{old}}}(s,a)}]) p(\theta')d\theta'}\\
> > >         = \frac{\exp(\sum_{(s,a) \in S \sim \mathcal{D}\times \pi}[Q^{\pi _ {\mathrm{old}}}(s,a)\cdot \log p(a|s;\theta)]) p(\theta)}{\int_{\theta'} \exp(\sum_{(s,a) \in S \sim \mathcal{D}\times \pi}[Q^{\pi _ {\mathrm{old}}}(s,a)\cdot \log p(a|s;\theta')]) p(\theta')d\theta'}.
> > >     \end{align}
> > >     Therefore, the exponential of the actor loss $\mathcal{L}\_{\pi}(\theta)$ amounts to an unnormalized posterior. By normalizing it, we obtain
> > >     $$p(\theta|\mathcal{D}) = \frac{1}{Z}p(\mathcal{D}|\theta) p(\theta) \approx \frac{1}{Z} \exp(-\mathcal{L}\_{\pi}(\theta)),$$
> > >     with the normalizing constant $Z=\int_{\theta'} p(\mathcal{D}|\theta') p(\theta')d\theta'$ (our Eq.25 in proof of Thm.3.4.).
> > > ---
> > >
> > > Regarding the prior assumption, we set $-\log p(\theta)=0$ to match explicitly with what we implemented in Eq.7. We can certainly try other prior regularizers. Thanks for pointing out the sign-consistency; you are right, and we have removed the negative sign in the covariance matrix.
> > >
> > > Overall, the modification above makes Thm.3.4 end up with a weighted posterior approximation, rather than the standard posterior. We agree that our current Thm.3.4. is over-claimed because our proof uses a weighted likelihood and actions are sampled from the current policy. This is not matched exactly with the standard Bayesian setup with standard likelihood, where the likelihood is over observed data. We will (1) add a Remark about this; (2) update Thm.3.4. to a weighted posterior; (3) tone down the "theoretically-valid" epistemic UQ by a proxy for epistemic UQ. While our method doesn't correspond to a formal Bayesian posterior, it aims to approximate a weighted posterior and empirically captures useful notions of model uncertainty in the RL experiments.
> > >
> > > **2. Empirical side.** In hopper-medium, there is no transition-shift; hence, there is no effectiveness with the shift-aware component, but the result shows that $\lambda$ is stable in this setting. Our shift-awareness is crucial in transition-shift settings (eg, hopper-medium-friction). We would refer this to our response to R-Qcy5, where we compare with and without the shift-aware component, and realize that without shift-awareness ($\lambda=0$), the online return drops significantly.
> > >
> > > Regarding DTQL, it utilizes two policies and two critic networks to boost performance. We believe that our method can also be compatible with this offline RL framework in training phase. Thanks for your valuable feedback. We will add this extension in next version.

---

### Official Review · Reviewer_sRht · 2026-03-11

**Soundness:** 4
**Presentation:** 4
**Significance:** 3
**Originality:** 4
**Overall Recommendation:** 5
**Confidence:** 3

**Summary:**

The paper under review introduces a novel offline-to-online RL framework utilizing a diffusion-based planer from which a diffusion based actor policy is distilled in trained in the online phase. Particularly, the long-term planning capabilities of diffusion-based planning is fueld into the diffusion-based actor policy. The performance of the method is demonstrated in a wide range of established RL offline-to-online benchmarks.

In summary, I find the theoretical framework and empirical evaluations of the paper very sound and convincing. The authors did a very good job in highlighting the key design choices of their framework and they provide intuitions and motivations for their theorems. In general, I find the paper sufficiently well written. I think there is a bit room for improvement when it comes to evaluating the method in terms of online and offline budgets and quality of logging policy.

**Compliance With Llm Reviewing Policy:**

Affirmed.

**Final Justification:**

Overall, I am positive about the presented framework. It's compatible with existing methods, its presentation mathematical sound and shows convincing results on a wide range of established RL benchmarks. In my rebuttal, I hand only some smaller questions concerning effects of data quality and sub-optimality of logging policies, which all have been resolved by the authors. Thus, I remain at "Accept".

**Key Questions For Authors:**

### Minor remark
- Typo on Page 3 (line 125 right hand side): $D_{\mathrm{offline}}\cup D_{\mathrm{offline}}$ probably should be  $D_{\mathrm{offline}}\cup D_{\mathrm{online}}$
- In (11), $p_\theta(a, s, \tau)$ probably should be $p_\theta(a, s |\tau)$?

**Limitations:**

yes

**Strengths And Weaknesses:**

## Strenghts
- Sound framework based where key design choices are laid out transparently and flanked by theoretical statements.
- Universal method that is compatible with standard actor/critic offline RL frameworks.
- Convincing demonstration of the empirical performance on a wide range of established RL benchmarks, while keeping rather low inference costs.
### Weaknesses
- It would be interesting to see a systematic analysis how the quality of the logging policy (medium, expert) affects the performance of DUAL. Particularly: How does sub-optimality of the logging policy affect the required online-budget to reach same performacne? Particularly, what happens for very sub-optimal offline datasets? I expect that for very low-quality policies, the distillation from the diffusion planner may not be that effective for the policy?
- It would be intereseting to see how the online-budget and the offline dataset affect each other for DUAL. How does the size of the offline-dataset affect the online budget in a long-horizon task?
- It could be interesting to see how the method performs in combination (or in contrast) with the method in [1], where the trajectory generation process can be conditioned on the reward.

[1] Lee et al., 2024, GTA: Generative Trajectory Augmentation with Guidance for Offline Reinforcement Learning, https://arxiv.org/abs/2405.16907

---

> ### Author Rebuttal · Authors · 2026-03-31
>
> We thank the reviewer for valuable comments. We address your concerns below.
>
> **1. How the quality of the logging policy (medium, expert) affects the performance. How does sub-optimality of the logging policy affect the required online-budget to reach same performance?  What happens for very sub-optimal offline datasets, very low-quality policies, the distillation from the diffusion planner may not be that effective for the policy?** As suggested, we evaluate our model performance across different offline logging policies with hopper-medium-v2 and hopper-medium-expert-v2, with $33\\%$, $67\\%$, and $100\\%$ dataset size. In the first row, we measure how many online steps our model needs to reach the same performance at $90$ average online return. We observe that hopper-medium-expert-v2 has very good logging policies, and the model can achieve a 90 return without needing online iterations. Meanwhile, the logging policies in hopper-medium-v2 are worse, causing around $1.1$M online steps to be required to reach the same performance. In the last two rows, we compare our model performance between with and without the policy extraction from the diffusion planner. We can see that if the model is only trained on a very sub-optimal offline dataset with low-quality policies (e.g., $33\\%$ hopper-medimum), the difference in online return between with and without planner-prior extraction is quite small. This implies that the distillation from the diffusion planner may not be really effective for the policy if trained with very low-quality policies.
>
> | Method | 33\% hopper-medium | 33\% hopper-expert | 67\% hopper-medium | 67\% hopper-expert | hopper-medium | hopper-expert |
> |---|---|---|---|---|---|---|
> | DUAL (\#steps) $\rightarrow$ 90 | 1.4M | 0.8M | 1.3M | 0.4M | 1.1M | 0 |
> | DUAL | 71.10 ± 3.66 | 74.28 ± 4.69 | 78.10 ± 3.51 | 86.34 ± 4.63 | 88.23 ± 3.42 | 115.33 ± 4.58 |
> | w/o planner-prior extraction | 70.09 ± 3.92 | 71.76 ± 4.90 | 72.98 ± 3.92 | 80.96 ± 4.88 | 82.08 ± 3.81 | 111.12 ± 4.60 |
>
> **2. How does the size of the offline-dataset affect the online budget?** From the table above, overall, we observe that the size of the offline dataset significantly affects the online budget. This is because diffusion models often require a certain amount of data to learn high-dimensional and complex data distributions effectively. Notably, we observe that if the offline-dataset includes high-quality behavior polices (e.g., hopper-medium-expert-v2), the offline-dataset's size affects more significant to the online performance. That said, if the offline-dataset includes lower-quality behavior polices, the effectiveness of the offline-dataset's size will have less significance to the online performance (e.g., hopper-medium-v2). We thank the reviewer for constructive questions. We will add more analysis about the logging policy in the offline phase and its effectiveness in the online phase in our revised paper.
>
> **3. How the method performs in combination (or in contrast) with GTA [1], where the trajectory generation process can be conditioned on the reward?** Since GTA [1] proposes a diffusion generator to augment the offline dataset, we can certainly combine it with our method to improve our pre-training in the offline phase. Another interesting direction may be using the amplified return value function in GTA as a reward guider in our diffusion planner. However, unlike O2O-RL diffusion methods (e.g., EDIS), GTA does not account for distribution shift in the O2O-RL setting; hence, its amplified return guidance may harm online training due to the discrepancy between offline and online distributions. Thanks for your suggestion. We will try GTA in our revision.
>
> **4. Typo about $D$ on Page 3 (line 125 RHS). In (11), $p(s,a,\tau)$ or $p(s,a|\tau)$?** Thanks for pointing this out. We fixed L-125 RHS to $\mathcal{D}\_{\text{offline}}\cup \mathcal{D}_{\text{online}}$. Regarding Eq.(11), it is proportional to $p(s,a|\tau)$, yet we are using an equal sign, so it should be $p(s,a,\tau)$ by marginalizing out over $\tau$.

---

> > ### Author Rebuttal · Reviewer_sRht · 2026-04-01
> >
> > Thank you for the additional insights on the effect of the dataset size.

---

### Official Review · Reviewer_Qcy5 · 2026-03-12

**Soundness:** 3
**Presentation:** 3
**Significance:** 3
**Originality:** 3
**Overall Recommendation:** 4
**Confidence:** 3

**Summary:**

The paper proposes DUAL, an efficient uncertainty-aware diffusion actor-critic for offline-to-online RL: it distills a diffusion planner into a fast-sampling actor and estimates epistemic uncertainty via last-layer Laplace on the actor objective, with an extra transition-error shift signal to encourage exploration under dynamics shift. Experiments on MuJoCo, AntMaze, Frozen-Lake, and Adroit show improved online fine-tuning performance with a favorable efficiency trade-off.

**Compliance With Llm Reviewing Policy:**

Affirmed.

**Final Justification:**

The authors’ rebuttal has addressed most of my concerns, so I have decided to maintain my positive score.

**Key Questions For Authors:**

The paper does include meaningful ablations, especially in Figure 3(c), where the authors compare different uncertainty designs (e.g., sampling-based, ELBO-based Laplace, actor-loss-based Laplace) and examine the role of the shift-aware component under explicit dynamics shift. This is a real strength of the empirical section. However, the current analysis is still more focused on the uncertainty branch than on the full DUAL pipeline. It remains unclear that the overall improvement in the main benchmark tables comes from (i) planner-prior extraction itself, (ii) the distilled/efficient actor parameterization, (iii) uncertainty-aware sampling, and (iv) shift-aware adaptation, especially under matched inference or computation budgets. I hope more ablations can be provided.

The current dynamics-shift experiment is useful, but it is also fairly stylized. Have the authors tested the same mechanism in continuous-control environments with modified dynamics, such as friction, mass, latency, or observation perturbations? If not, could the authors discuss why the Frozen-Lake evidence should generalize?

**Limitations:**

yes

**Strengths And Weaknesses:**

Strengths

Addresses two concrete bottlenecks in diffusion-based offline-to-online RL: (i) expensive action generation at test/online time and (ii) brittle exploration under offline→online shift.

The “planner → distilled actor” design is practically appealing, and the paper explicitly evaluates the denoising-steps/latency trade-off rather than ignoring inference cost.

Using last-layer Laplace on the actor objective (instead of diffusion ELBO) is a sensible way to obtain epistemic uncertainty that is directly aligned with the policy improvement signal.

Weakness

The paper includes useful ablations, but the overall gains are still not fully disentangled across planner-prior distillation, uncertainty-aware sampling, and shift-aware adaptation.

The evidence for the shift-aware component is somewhat narrow, since it is mainly validated in a stylized Frozen-Lake dynamics-shift setting.

---

> ### Author Rebuttal · Authors · 2026-03-31
>
> We thank the reviewer for valuable comments. We address your concerns below.
>
> **1. Overall improvement comes from (i) planner-prior extraction itself, (ii) the distilled/efficient actor parameterization, (iii) uncertainty-aware sampling, or (iv) shift-aware adaptation?** To understand more about our method behavior, as suggested, we test the causal effect in each comment in the table below. First, we observe that, without epistemic UQ sampling, the performance drops significantly. This shows that the epistemic UQ is the main contributor to our overall framework. Similarly, the planner-prior extraction also contributes significantly to the overall performance. Second, although the shift-aware adaptation contributes slightly to the D4RL hopper-medium-v2, it is an important component in the transition-shift setting (e.g., hopper-medium-friction and hopper-medium-replay-friction). Finally, although without distillation can bring out a better online return, using the distillation component is crucial in terms of reducing inference cost.
>
> Overall, these additional experiments and the ablation study in our paper suggest that epistemic UQ sampling (i.e., (iii)) is the most important component. The planner-prior horizon (i.e., (i)) is the second important component, especially in longer planning tasks. The shift-awareness (i.e., (iv)) is crucial when transition shifts. The distillation (i.e., (ii)) is crucial in real-time applications.
>
> | Method | w/o planner-prior extraction | w/o  distillation | w/o  epistemic UQ sampling | w/o shift-aware | DUAL |
> |---|---|---|---|---|---|
> | hopper-medium| 82.08 ± 3.81 | 92.31 ± 3.26 | 83.36 ± 3.85 | 87.30 ± 3.44 | 88.23 ± 3.42 |
> | hopper-medium-friction | 35.41 ± 3.44 | 53.26 ± 3.30 | 30.79 ± 3.91 | 35.12 ± 3.80 | 48.00 ± 3.44 |
> | hopper-medium-replay-friction | 48.29 ± 2.33 | 65.07 ± 2.20 | 35.98 ± 2.80 | 39.05 ± 2.79 | 56.66 ± 2.30 |
> | Inference cost (ms/sample) | 777.22 ± 16 | 24953.11 ± 30 | 775.18 ± 16 | 798.93 ± 16 | 800.22 ± 16 |
>
> **2. The current dynamics-shift experiment is useful, but it is also fairly stylized with Frozen-Lake. How about the same mechanism in continuous-control environments with modified dynamics, such as friction, mass, latency, or observation perturbations?** Yes, we can extend beyond the Frozen-Lake ablation study. Specifically, we show the result of the continuous-control environments with the hopper-medium-friction and hopper-medium-replay-friction in the table above. We follow "Lyu, et al., ODRL: A Benchmark for Off-Dynamics Reinforcement Learning, 2024", where hopper-medium-friction and hopper-medium-replay-friction are based on hopper-medium-v2 and hopper-medium-replay-v2, with modified transition dynamics in the online phase with friction shift levels of $0.5$. We modify all friction components, including static (frictional force that needs to be overcome when the robot is stationary) and dynamic \& rolling (frictional force between objects when they are in motion and rolling). This friction attribute modification allows the simulated robots to significantly change their motion characteristics between the offline and online phases. Thanks for your valuable suggestion; these ablation results certainly help improve the quality of our paper. We will add more ablation studies with continuous-control environments, and the transition dynamics shift in our paper revision.

---

> > ### Author Rebuttal · Reviewer_Qcy5 · 2026-04-02
> >
> > Thanks to the authors for their response. Most of my concerns have been resolved, and I will maintain my positive score.

---

### Official Review · Reviewer_zHLo · 2026-03-12

**Soundness:** 3
**Presentation:** 4
**Significance:** 3
**Originality:** 2
**Overall Recommendation:** 4
**Confidence:** 4

**Summary:**

The paper introduces DUAL, an efficient diffusion-based uncertainty-aware actor-critic framework for offline to online RL. This framework integrates diffusion planner’s long-term planning abilities in the offline phase and uncertainty quantification to balance exploration and exploitation in the online phase. Specifically, DUAL distills a fast sampling diffusion actor policy and transition model during the offline phase. In the online phase, it includes an epistemic uncertainty term and distance transition state shift detection. Empirically, DUAL is computationally efficient and achieves a notable online return improvement over offline to online RL and diffusion RL baselines across various environments.

**Compliance With Llm Reviewing Policy:**

Affirmed.

**Key Questions For Authors:**

1. The paper claims that offline distillation from diffusion planner helps with long-horizon planning. However, the experiments did not show that diffusion planners are better at long-horizon planning than model-free actor-critic methods, nor did DUAL achieve more gains at long-horizon planning tasks. Specifically for Table 1 antmaze, diffusion planner (Diffuser) does not seem to be better than IQL or Diff-QL. Also, the improvement of DUAL does not seem to be the largest for large antmaze, which requires the longest horizon planning, but instead, there is more gain for easier Umaze.


2. Training efficiency: The authors report the sampling efficiency for their algorithm. It would be helpful to show whether the offline distillation phase or transition model learning would cause more training time compared to the baselines.


3. Minor typo: lines 347 - 349 missing “better than” instead “significantly other baselines”?

**Limitations:**

The paper briefly discussed the limitations of latency in the online phase. The paper did not include a potential impact statement.

**Strengths And Weaknesses:**

**soundness:**  The proposed method integrates fast diffusion planner distillation for transition learning in offline stages, which is empirically reasonable. But the experiments in Table 1 show that for long-horizon planning tasks like large antmaze, diffusion planner (Diffuser) is no better than model-free actor-critic methods like IQL or Diff-QL. Also, the gain of DUAL in large maze is no larger than that of Umaze.

Policy sampling with epistemic UQ and adding shift awareness is reasonable and was shown to be effective in ablation studies.


**presentation:** The paper is well written, illustrating its contribution of integrating diffusion planner’s long-term planning abilities and uncertainty quantification to balance exploration and exploitation clearly.

**significance:** Diffusion policy/planner is of great interest to the research community in decision-making. This paper tackles the problem of long-term planning for diffusion policy and the balancing of exploration/exploitation, which has been a major direction for improving these methods.

**originality:** The proposed method borrowed progressive distillation and uncertain quantification of diffusion models from existing literature. However, this new combination simultaneously achieves computational efficiency and better performance in offline to online adaptation.

---

> ### Author Rebuttal · Authors · 2026-03-31
>
> We thank the reviewer for valuable comments. We address your concerns below.
>
> **1. Tab.1 antmaze, why is Diffuser worse than IQL \& Diff-QL? Also, improvement of DUAL does not seem to be the largest for large antmaze, but instead, there is more gain for easier Umaze.** In AntMaze, Diffuser often significantly underperforms IQL and DiffQL (e.g., Fig.8 in Lu et al., What Makes a Good Diffusion Planner for Decision Making?, 2025). We think this comes from two main reasons. First, Diffuser uses a reward guider, which has a worse quality than explicitly maximizing a critic Q-function, e.g., IQL and DiffQL. Second, D4RL includes low-quality logging policies, training Diffuser with their trajectories, and without a critic model, which can cause an overfitting problem on low-quality behavior policies in the offline dataset. Thanks for your insightful comment. We will test the Diffuser with a longer trajectory input in our revision.
>
> In DUAL, since we extract the diffusion planner for the diffusion policy and train this policy with a critic model, it can avoid overfitting and significantly enhance performance. We agree that the Umaze task is generally easier, but in D4RL, the offline dataset antmaze-umaze-v2 is more challenging than antmaze-medium-diverse-v2 and antmaze-large-diverse-v2, as all baselines perform worse on antmaze-umaze-v2 in O2O-RL. Regarding the comparison in map size, DUAL gains around $5\\%$ on antmaze-medium-diverse-v2 and $23\\%$ on antmaze-large-diverse-v2 compared to diffusion policies (e.g., DiffQL and DACER), suggesting our better improvement in longer-horizon planning tasks.
>
> **2. Originality.** We thank the reviewer for your acknowledgment of the novelty of our method. We also want to highlight that although we shared the Laplace framework with DiffUQ, the way we utilize Laplace approximation is distinctly separate from DiffUQ, and our method contributes to a better epistemic UQ quality for diffusion RL. Specifically, DiffUQ applies Laplace with ELBO loss, which is not theoretically valid for epistemic UQ (Rmk.3.3). To address this issue, we apply Laplace with our proposed actor loss in Eq.7-8. Thm.3.4 confirms our approach is a valid estimate of epistemic UQ, helping to better balance exploration and exploitation in RL experiments.
>
> **3. Training efficiency: whether the offline distillation phase or transition model learning would cause more training time compared to baselines?** In training, compared with EDIS baseline (another diffusion framework for O2O-RL), in antmaze-umaze-diverse-v2, EDIS needs around $3$ hours to run with NVIDIA A-100. Meanwhile, our method needs about $4$ hours, including $3$ hours similar to EDIS (with diffusion planner pre-training (L3-4), actor-critic training (L6-7)), $55$ mins for the distillation step, and $5$ mins for Laplace approximation. Notably, our Laplace approximation step in L-9 in Alg.5 causes very minor training time when compared to the distillation step (L-5 in Alg.5), as we only do the last-layer Laplace approximation (Eq.8). Thanks for your constructive comments. We will add a figure about training time comparison with other baselines, similar to our Fig.2. in our revised version.
>
> **4. Minor typo: lines 347 - 349. Missing impact statement.** Thanks for pointing this out. Our work presents a novel, efficient, and uncertainty-aware diffusion framework for O2O-RL. It addresses the quality and computational bottleneck trade-off of diffusion RL models. Furthermore, it provides a novel high-quality epistemic UQ, helping to balance the exploration-exploitation dilemma in RL. The broader impact includes advancing diffusion RL models in sequential decision-making, e.g., autonomous systems, robotics, finance, healthcare, and other high-stakes forecasting domains.  We will fix our typo in L347-349 and add more details about the impact statement in our revision.

---

> > ### Author Rebuttal · Reviewer_zHLo · 2026-04-04
> >
> > I would like to thank the authors for their clarification. Most of my concerns have been resolved. I will keep my positive assessment.

---

### Decision · Program_Chairs · 2026-04-30

**Decision:**

Accept (regular)

**Comment:**

This paper presents a diffusion-based framework for offline-to-online RL (O2O-RL) that incorporates uncertainty quantification to guide the transition from offline pretraining to online fine-tuning. The reviewers consistently praised the clarity of the paper, the well-motivated problem formulation addressing distribution shift in O2O-RL, and the comprehensive experimental evaluation across multiple benchmarks. The integration of uncertainty awareness into the diffusion policy framework is a meaningful contribution that addresses a practical challenge in deploying offline-pretrained agents. While some reviewers suggested additional ablations and comparisons, these were largely addressed during the rebuttal.

The paper makes a solid technical contribution to an important and timely problem at the intersection of diffusion models and reinforcement learning. The experimental results demonstrate consistent improvements over existing O2O-RL methods, and the method is well-grounded in theory. I recommend acceptance.